# Investigating the physical mechanisms that modify wind plant blockage in stable boundary layers

Miguel Sanchez Gomez[1], Julie K. Lundquist[1,2,3], Jeffrey D. Mirocha[4], and Robert S. Arthur[4]

[1]Department of Atmospheric and Oceanic Sciences, University of Colorado Boulder, Boulder, Colorado, 80309-0311, United States
[2]National Renewable Energy Laboratory, Golden, Colorado, 80401, United States
[3]Renewable and Sustainable Energy Institute, Boulder, Colorado, 80309, United States
[4]Lawrence Livermore National Laboratory, Livermore, California 94550, United States

**Correspondence:** Miguel Sanchez Gomez (misa5952@colorado.edu)

**Abstract.** Wind plants slow down the approaching wind, a phenomenon known as blockage. Wind plant blockage undermines turbine performance for front-row turbines and potentially for turbines deeper into the array. We use large-eddy simulations to characterize blockage upstream of a finite-size wind plant in flat terrain for different atmospheric stability conditions, and investigate the physical mechanisms modifying the flow upstream of the turbines. To examine the influence of atmospheric stability, we compare simulations of two stably stratified boundary layers using the Weather Research and Forecasting model in large-eddy simulation mode, representing wind turbines using the generalized actuator disk approach. For a wind plant, a faster cooling rate at the surface, which produces stronger stably stratified flow in the boundary layer, amplifies blockage. As a novelty, we investigate the physical mechanisms amplifying blockage by evaluating the different terms in the momentum conservation equation within the turbine rotor layer. The velocity deceleration upstream of a wind plant is caused by an adverse pressure gradient and momentum advection out of the turbine rotor layer. The cumulative deceleration of the flow upstream of the front-row turbines instigates vertical motions. The horizontal flow is diverted vertically, reducing momentum availability in the turbine rotor layer. Although the adverse pressure gradient upstream of the wind plant remains unchanged with atmospheric stability, vertical advection of horizontal momentum is amplified in the more strongly stable boundary layer, mainly by larger shear of the horizontal velocity, thus increasing the blockage effect.

## 1 Introduction

Wind turbines and wind plants each modify the flow within their vicinity. The flow downstream (i.e., the wake) is slower and more turbulent as turbines extract kinetic energy from the wind. The wind also decelerates upstream of a wind plant, an effect known as blockage (Bleeg et al., 2018). This region of slower wind speed is called the induction region of the wind plant. Wind turbine and wind plant wakes can also affect power production of downstream turbines (El-Asha et al., 2017) and plants (Stieren and Stevens, 2022), an effect known as wake loss. Wake losses are accounted for in energy-production loss estimates (Filippelli et al., 2018) and are of the order of 10% of the annual energy production (AEP) in wind plants (Lee and Fields, 2021). Blockage also reduces power production in wind plants (Bleeg et al., 2018; Sebastiani et al., 2021). However,

blockage is usually neglected in loss estimates due to uncertainty in the magnitude of this effect, possibly resulting in lower-than-forecasted energy production and financial losses for wind plant operators (Brower, 2012; Bleeg et al., 2018; Ørsted, 2019; Lee and Fields, 2021).

There is disagreement on the magnitude of the velocity deceleration within the induction region of wind plants. The velocity deceleration within the induction region can vary substantially depending on the size and layout of the wind plant (e.g., Centurelli et al., 2021; Strickland and Stevens, 2022; Bleeg et al., 2018; Strickland et al., 2022), atmospheric conditions (e.g., Allaerts and Meyers, 2017, 2018; Bleeg and Montavon, 2022; Schneemann et al., 2021; Strickland et al., 2022), wind turbine characteristics (e.g., Ebenhoch et al., 2017), and wind speed (e.g., Schneemann et al., 2021). A limited set of simulations report changes in hub-height wind speed larger than 10% at a distance of 2 rotor diameters (2D) upstream of the first row of turbines (Allaerts and Meyers, 2017; Wu and Porté-Agel, 2017; Allaerts and Meyers, 2018). Conversely, experimental studies and numerical simulations report between 1% and 5% changes in hub-height wind speed at a distance of 1.5D to 3D upstream of the first row of turbines (Bleeg et al., 2018; Bleeg and Montavon, 2022; Schneemann et al., 2021; Centurelli et al., 2021; Segalini and Dahlberg, 2020; Jacquet et al., 2022; Strickland et al., 2022). The large spread in the magnitude of the blockage effect suggests there may be multiple physical mechanisms modifying the velocity deceleration in the induction region.

Idealized simulations of large wind plants suggest gravity waves could potentially amplify blockage (Wu and Porté-Agel, 2017; Allaerts and Meyers, 2017, 2018; Maas, 2023). Large-eddy simulations (LES) by Wu and Porté-Agel (2017) show increasing stratification in the troposphere can result in upstream propagating gravity waves. Gravity waves propagate upstream in their strong free-atmosphere stratification simulation but do not propagate upstream in their weak free-atmosphere stratification simulation (Wu and Porté-Agel, 2017). Stronger surface layer stability, longer wind plants, and shallower boundary layers may also increase gravity wave-amplified blockage (Allaerts and Meyers, 2018, 2017; Maas, 2023). Note that Allaerts and Meyers (2017, 2018) and Maas (2023) simulate the flow around an infinitely wide wind plant. The power loss due to upstream-propagating gravity waves increases as the wind plant becomes infinitely wide (Allaerts and Meyers, 2019). Therefore, the velocity deceleration in the induction region of an infinitely wide wind plant is likely larger than would be expected in an operational wind plant of finite width. Wind plants approach the infinitely-wide regime when they are two orders of magnitude wider than the boundary layer height (Allaerts and Meyers, 2019); hence, a 100 km wide wind plant can be considered infinitely wide for a 1 km deep boundary layer. Even though gravity waves in idealized simulations can potentially produce $\mathcal{O}(10\%)$ velocity reductions in the induction region ($x < -2D$), these large decelerations have not yet been observed in operational wind plants.

The nature of the physical mechanism modifying blockage with minimal upstream propagation of gravity waves, where the velocity slowdown is on the order of 1-5% of freestream (Bleeg et al., 2018; Segalini and Dahlberg, 2020; Centurelli et al., 2021; Schneemann et al., 2021; Bleeg and Montavon, 2022; Jacquet et al., 2022), has not been studied in depth. Atmospheric conditions have been shown to modify blockage, though the mechanisms through which this happens are unclear. Using a scanning lidar, Schneemann et al. (2021) measured $4\% \pm 2\%$ velocity reduction between 30D and 5D upstream of the first turbine row during stable surface conditions. They did not observe blockage during unstable surface conditions (Schneemann et al., 2021). Similarly, Bleeg and Montavon (2022) show larger power reductions for a single turbine row with increasing

surface-layer and free-atmosphere stratification. Simulating an infinitely wide wind plant, Strickland et al. (2022) also demonstrate increased blockage with stronger surface layer stability. They propose blockage in stable conditions is amplified by a cold air anomaly that produces a high pressure region upstream of the wind plant (Strickland et al., 2022). However, they do not quantify the increase in horizontal pressure gradient caused by the cold air anomaly nor its relative importance compared with other forcing mechanisms (Strickland et al., 2022). Wind plant layout also influences the velocity deceleration in the induction region. Using LES of neutrally stratified boundary-layer flow, Strickland and Stevens (2022) show an increase in the adverse pressure gradient upstream of wind plants with closely-spaced turbines in the cross-stream direction. Bleeg and Montavon (2022) also show blockage varies between an infinitely-wide and finite-sized wind plant. The majority of studies on blockage focus on the parameters (e.g., turbine layout, atmospheric conditions) that modify blockage but not the physical mechanisms that influence the flow.

Here, we investigate how atmospheric stability modifies upstream blockage with minimal upstream propagation of gravity waves (see Appendix C for a discussion on gravity waves in our domain). Specifically, we investigate: (1) if the wind speed deceleration upstream of a finite-sized wind plant is modified with atmospheric stability, and (2) the physical mechanisms that modify the blockage effect in stably stratified flow. To investigate the impact from atmospheric stability, we simulate the flow around a wind plant for two distinct stable boundary layers. Furthermore, we run a set of simulations for a stand-alone turbine for each atmospheric condition to establish a baseline blockage effect for comparison.

This paper is structured as follows. We describe our simulation setup and stability cases in Sect. 2. We show how the velocity field in the induction region is modified by atmospheric static stability in Sect. 3. In Sect. 4, we analyze the physical mechanisms modifying wind plant blockage. Finally, in Sect. 5, we discuss our findings and provide suggestions for future work that could further improve understanding of the wind plant blockage effect.

## 2 Methodology

### 2.1 Large-eddy simulation setup

We perform LES of a wind plant under stably stratified conditions using the Weather Research and Forecasting (WRF) model v4.1.5 (Skamarock et al., 2019) with turbines represented using a generalized actuator disk (GAD) approach (Mirocha et al., 2014; Aitken et al., 2014; Arthur et al., 2020). WRF is a fully compressible, nonhydrostatic model that solves the Navier-Stokes and thermodynamic equations for large-Reynolds number fluids (no viscosity or thermal conductivity). WRF uses an Arakawa-C grid staggering in the horizontal and a hydrostatic pressure-based vertical coordinate. Equations are integrated in time using a 3rd-order Runge-Kutta scheme with a smaller time step for acoustic modes. The advection terms are spatially discretized using a hybrid 5th- and 3rd-order scheme in the horizontal and vertical directions, respectively, which improves the model's effective resolution to $4-5\Delta x$ (Kosović et al., 2016).

We use a two-domain configuration with flat terrain to evaluate the blockage effect from wind plants. A periodic LES domain provides the boundary conditions for a nested LES domain via one-way nesting (i.e., atmospheric conditions for the outermost grid cells in the nested domain are specified from the parent domain). Horizontal and vertical grid spacing remains the same

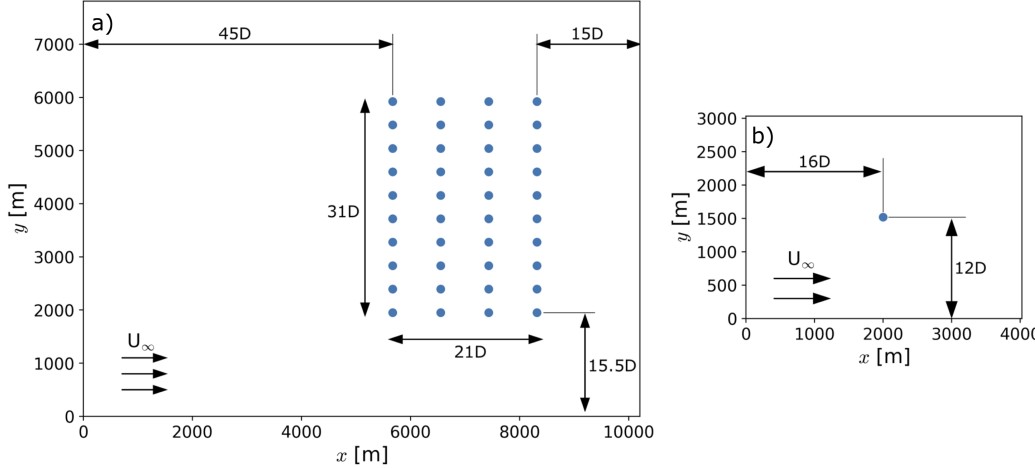

**Figure 1.** Relative location of the turbines in the wind plant (a) and stand-alone turbine (b) simulations for evaluating blockage. Forty NREL 5MW wind turbines constitute the 200MW wind plant simulated herein. Turbine spacing is 7D and 3.5D in the streamwise and cross-stream directions, respectively.

for the parent and nest domains. We use the same domain characteristics, but a smaller nested domain size, to evaluate the blockage effect for a stand-alone wind turbine. We simulate moderately and weakly stably stratified flow. Because turbulence structures vary with atmospheric stability (Wurps et al., 2020), we use a finer grid in the more stable case (see Appendix D for a discussion on grid resolution). Horizontal grid spacing is $\Delta x = 5$ m and $\Delta x = 7$ m for the moderate and weak stability conditions, respectively. Similarly, vertical grid spacing at the surface is $\Delta z_s = 5$ m in the moderate stability case and $\Delta z_s = 6$ m in the weak stability case. The parent domain is 10 grid points larger than the nest in the horizontal directions. A summary of the LES domains is in Table 1.

**Table 1.** Simulation setup, including stability case, surface cooling rate $\dot{T}_s$, turbine layout, domain size $(L_x, L_y, L_z)$, horizontal resolution $(\Delta x, \Delta y)$, vertical resolution at the surface $(\Delta z_s)$, and non-staggered grid points $(n_x, n_y, n_z)$.

| Case | $\dot{T}_s$ [K h$^{-1}$] | Turbine layout | $L_x, L_y, L_z$ [km] | $\Delta x, \Delta y$ [m] | $\Delta z_s$ [m] | $n_x, n_y, n_z$ |
|---|---|---|---|---|---|---|
| Moderate stability | -0.5 | Wind Plant | 10.3, 7.8, 3.5 | 5 | 5 | 2062, 1560, 105 |
| | | Stand-alone turbine | 4.03, 3.04, 3.5 | | | 806, 607, 105 |
| Weak stability | -0.2 | Wind plant | 10.3, 7.8, 3.5 | 7 | 6 | 1470, 1115, 66 |
| | | Stand-alone turbine | 4.03, 3.04, 3.5 | | | 576, 434, 66 |

All simulations are initialized with a dry atmosphere and zero latent heat flux. No cloud, radiation, or land surface models are used in the LES domains. An implicit Rayleigh damping term is applied to the vertical velocity in the upper $1000\,\mathrm{m}$ of each domain to avoid wave reflection from the model top (Klemp et al., 2008). Surface boundary conditions are specified using Monin-Obukhov similarity theory with a surface roughness of $z_0 = 0.1\,\mathrm{m}$. We use the nonlinear backscatter and anisotropy (NBA) model with turbulence kinetic energy (TKE)-based stress terms (Kosović, 1997; Mirocha et al., 2010) to parameterize subgridscale (SGS) fluxes of momentum and heat.

Turbines are simulated exclusively in the nested domain using the generalized actuator disk implemented in WRF by Mirocha et al. (2014) and modified by Aitken et al. (2014) and Arthur et al. (2020). The NREL 5MW wind turbine has a hub-height of $90\,\mathrm{m}$, a rotor diameter $D$ of $126\,\mathrm{m}$, cut-in speed at $3\,\mathrm{m\,s^{-1}}$, rated speed at $11.4\,\mathrm{m\,s^{-1}}$, and cut-out speed at $25\,\mathrm{m\,s^{-1}}$ (Jonkman et al., 2009). Both the wind plant and stand-alone turbine layouts used in our simulations are shown in Figure 1. As will be shown later in the manuscript, the velocity deceleration in the induction region is virtually zero 30D upstream of the wind plant. Therefore, 45D of fetch upstream of the wind plant is deemed sufficient to investigate the induction region of the turbine array. Strickland and Stevens (2022) show the power of front-row turbines in a wind plant is sensitive to the ratio between the wind plant width ($L_{y-wp}$) and the domain size in the $y-$direction ($L_y$). Because the change in turbine power for ratios $L_{y-wp}/L_y < 0.5$ is small (Strickland and Stevens, 2022) but the increase in computational resources is significant, we use a ratio of 0.5 here. Our wind plant has an aspect ratio of $\sim 3:2$ to amplify the blockage effect as suggested by Allaerts and Meyers (2019). Segalini and Dahlberg (2020) found the blockage effect remains nearly constant when the wind plant has three or more rows, thus we include four turbine rows in our plant. Further, we constrain wind turbine spacing to 7D and 3.5D in the streamwise and cross-stream directions, respectively, following typical spacing in actual wind plants (Stevens et al., 2017).

Note that throughout the manuscript we denote temporal averaging using an overbar $\left(\overline{\phantom{x}}\right)$, spatial averaging along the $i-$direction using angled brackets $\langle\ \rangle_i$, and a normalized quantity using a hat $(\hat{\phantom{x}})$.

## 2.2 Atmospheric conditions

We simulate two different boundary layers to evaluate how blockage varies with atmospheric stability. Distinct stability conditions are obtained by providing different forcing at the surface. As suggested by Basu et al. (2008), we prescribe a temporal cooling rate rather than a heat flux at the surface. Moderately and weakly stably stratified flows are obtained by forcing the boundary layer with $\dot{T}_s = -0.5\,\mathrm{K\,h^{-1}}$ and $-0.2\,\mathrm{K\,h^{-1}}$, respectively.

A fully developed stable boundary layer is attained by spinning up a turbulent, neutral boundary layer, then adding a cooling rate at the surface. We initialize our simulations with a uniform potential temperature profile of $\theta = 300$ K up to $z = 1000$ m, a capping inversion from $1000\,\mathrm{m} < z < 1200\,\mathrm{m}$ with $\partial\theta/\partial z = 0.01\,\mathrm{K\,m^{-1}}$, and we specify $\partial\theta/\partial z = 0.001\,\mathrm{K\,m^{-1}}$ in the troposphere aloft. Both simulations are initialized with a $U_g = 11\,\mathrm{m\,s^{-1}}$ geostrophic wind speed and the Coriolis parameter is $f_c \approx 9.37 \times 10^{-5}\,\mathrm{s^{-1}}$, corresponding to a latitude of $40°$. Furthermore, we speed up turbulence development by adding $\pm 0.5$ K perturbations to the potential temperature field below the capping inversion at initialization. To reduce computational requirements, we spin-up turbulence and atmospheric stability in a small, precursor domain. After the flow is fully turbulent

and stably stratified, we tile multiple precursor domains along the $x-$ and $y-$directions to form a large domain. A complete description of this tiling methodology is presented in Appendix A.

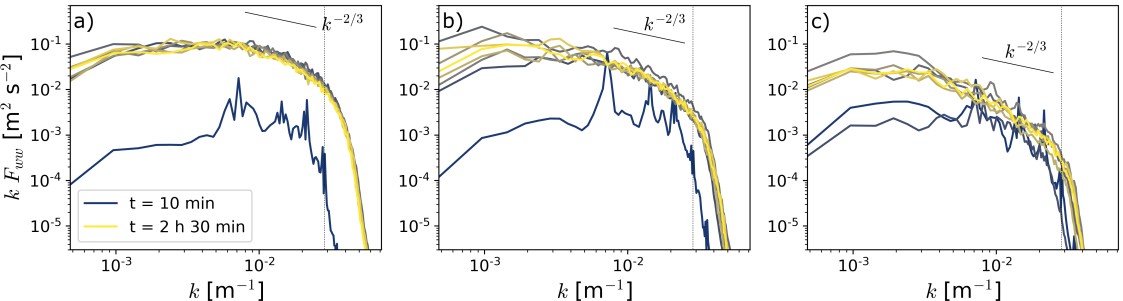

**Figure 2.** Compensated turbulence spectra of the $w-$velocity for the $\Delta x = 7\,\mathrm{m}$ neutrally stratified boundary layer in the precursor simulation at $z = 90\,\mathrm{m}$ (a), $z = 300\,\mathrm{m}$ (b), and $z = 800\,\mathrm{m}$ (c). Colored lines indicate time since initialization in 20-minute time increments. The dotted, black vertical line in each plot represents the effective grid resolution $(4-5\Delta x)$ expected from the reduced advection scheme in our simulations (Kosović et al., 2016). The theoretical -2/3 Kolmogorov slope for the inertial range is indicated by the solid black line in each plot.

A realistically turbulent neutral boundary layer develops shortly after initialization. Localized shear instabilities instigate turbulence throughout the boundary layer within the first hour of the simulation. These structures break up rapidly into smaller eddies, reducing shear until a quasi-steady state is reached. Turbulence structures form rapidly close to the surface and propagate upwards (Figure 2). At hub height, turbulence propagates across all resolvable scales after $20\,\mathrm{min}$ (Figure 2a). Further aloft, large- and small-scale turbulent motions develop after $30\,\mathrm{min}$ (Figure 2b). Turbulence propagates up to the capping inversion after $50\,\mathrm{min}$ (Figure 2c). Even though the flow is fully turbulent $1\,\mathrm{h}$ after initialization, turbulence and the mean flow in the boundary layer stabilize after $2\,\mathrm{h}$.

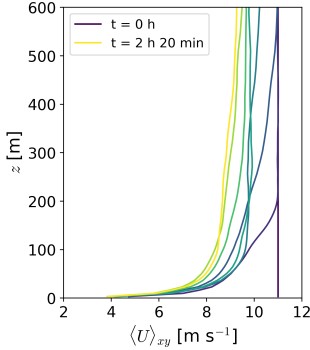

**Figure 3.** Vertical profile of the horizontal wind speed for the $\Delta x = 7\,\mathrm{m}$ neutrally stratified boundary layer in the precursor simulation. The velocity profile is averaged spatially over the entire domain. Colored lines indicate time since initialization in 20-minute time increments.

Spin-up for the neutral boundary layer is complete when changes in the mean flow and in turbulence statistics within the boundary layer are small over time. The spatially averaged horizontal velocity changes less than 1% in the boundary layer 2 h after initialization (Figure 3). Turbulence production at large scales from shear stabilizes after $1.5\,\mathrm{h}$ as the mean flow approaches equilibrium (Figure 2). Two hours after initialization, turbulent momentum transport at the surface reaches a quasi-steady state, turbulence spectra in the entire boundary layer converge, and changes in the streamwise velocity below the capping inversion

are small. At this point, we prescribe a cooling rate at the surface.

Boundary layer evolution varies with surface forcing (Figure 4). A fast cooling rate (i.e., $-0.5\,\mathrm{K\,h^{-1}}$) produces increasing temperature stratification below $400\,\mathrm{m}$ and quasi-neutral stratification up to the capping inversion (Figure 4c). The rapid development of a stable layer close to the surface reduces the vertical transport of momentum, suppressing turbulence aloft. A broad low-level jet (LLJ) develops after $4\,\mathrm{h}$ as turbulence aloft decreases (Figure 4a,b). Boundary-layer evolution for the slower

cooling rate is slightly different. A $\dot{T}_s = -0.2\,\mathrm{K\,h^{-1}}$ produces increasing temperature stratification up to the capping inversion (Figure 4h). The slow cooling rate initially produces nearly uniform cooling of the entire turbulent layer. After $3\,\mathrm{h}$, temperature stratification close to the surface is large enough to reduce the vertical transport of momentum and a LLJ starts forming close to the capping inversion (Figure 4f,g). Because of a slower cooling rate, the gradual reduction in vertical turbulent mixing results in a deeper boundary layer in the weak stability case compared to the moderate stability case.

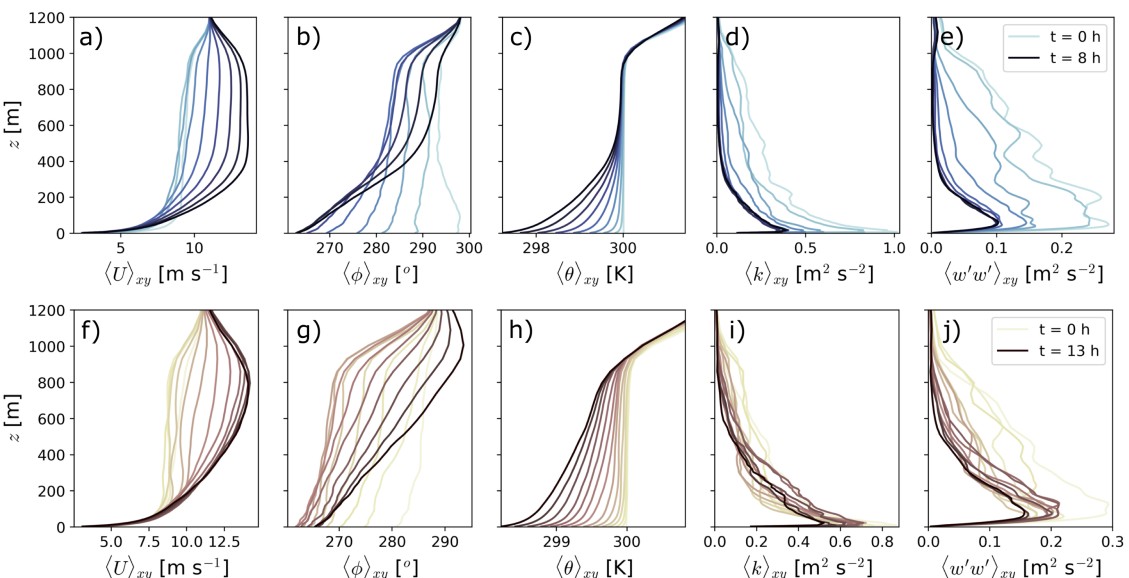

**Figure 4.** Evolution of atmospheric variables for the moderate (a-e) and weak (f-j) stability cases in the precursor simulations. Profiles are color-coded in $1\,\mathrm{h}$ increments.

Spin-up time varies for each stable simulation. The $-0.5\,\mathrm{K\,h^{-1}}$ simulation is run until the temporal change in bulk Richardson number in the surface layer and Obukhov length is small (Figure 5). The Obukhov length stabilizes to $L = 35\,\mathrm{m}$ after $8\,\mathrm{h}$; the bulk Richardson number between $z = 10\,\mathrm{m}$ and $z = 153\,\mathrm{m}$ stabilizes to $Ri_{bulk} = 0.14$. The nose of the LLJ after $8\,\mathrm{h}$ is

at $z = 363\,\mathrm{m}$ for the $-0.5\,\mathrm{K\,h^{-1}}$ case. The $-0.2\,\mathrm{K\,h^{-1}}$ simulation is run for 13 h. Even though $L$ and $Ri_{bulk}$ do not reach a quasi-steady state (Figure 5), the flow displays characteristics typical of stable boundary layers and stability metrics suggest surface layer stability is different from the $-0.5\,\mathrm{K\,h^{-1}}$ simulation. After 13 h, the Obukhov length is $L = 82\,\mathrm{m}$ and the bulk Richardson number is $Ri_{bulk} = 0.11$. After 13 h of simulation, the nose of the LLJ is at $z = 660\,\mathrm{m}$ for the $-0.2\,\mathrm{K\,h^{-1}}$ case. Note that we do not expect our simulations to reach a steady state because the cooling rate at the surface continually modifies stability in the surface layer. Nonetheless, the evolution of the surface layer is slow after 8 h (13 h) for the moderate (weak) stability simulation.

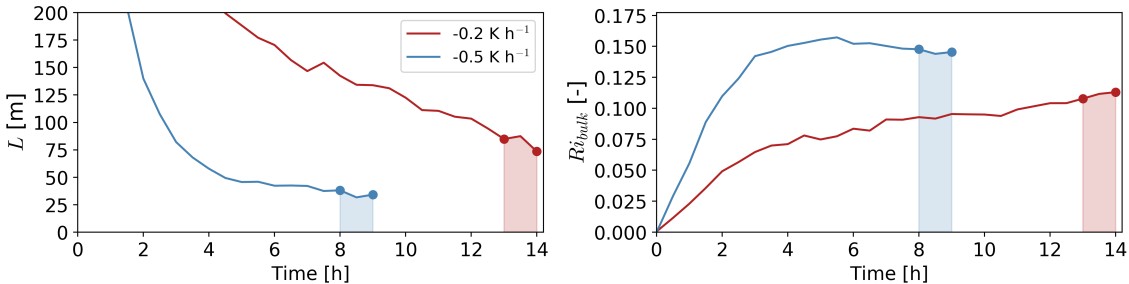

**Figure 5.** Temporal evolution of the Monin-Obukhov length (a) and bulk Richardson number (b) for each atmospheric state after prescribing a cooling rate at the surface. The bulk Richardson number is estimated between $z = 10\,\mathrm{m}$ and $z = 153\,\mathrm{m}$. The shaded colored areas in each plot represent the simulation time for evaluating blockage.

Atmospheric conditions after spin-up of turbulence and stability are shown in Figure 6. Hub-height wind speed and direction after spin-up are comparable for both atmospheric conditions (Figure 6a,b). For the moderate stability case, hub-height wind speed and direction are on average $8.9\,\mathrm{m\,s^{-1}}$ and $269.8°$, respectively; for the weak stability case, $8.3\,\mathrm{m\,s^{-1}}$ and $269.6°$, respectively. Over the simulation period, hub-height wind direction varies by less than $1°$ for both atmospheric conditions. The winds at hub height are in Region II of the power curve for the NREL 5MW turbine (Jonkman et al., 2009).

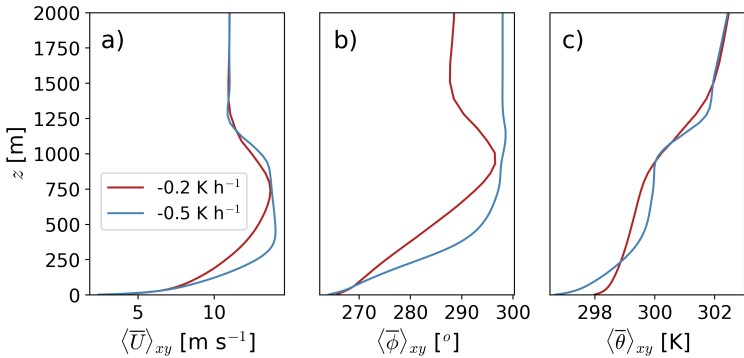

**Figure 6.** Horizontal wind speed (a), wind direction (b) and potential temperature (c) profiles for the atmospheric conditions simulated herein. Atmospheric variables are averaged spatially over the entire domain and temporally over 1 h after spin-up is complete.

Each stability case is run for 1 h, from which the first five minutes are discarded to guarantee that the flow at hub-height moves over the entire wind plant. The three-dimensional velocity, pressure, and potential temperature fields are output every 30 seconds, then time-averaged over the simulated time period.

## 3   The induction region

The three-dimensional, time-averaged velocity fields are averaged spatially to characterize the induction region. The velocity
field is averaged vertically over the turbine rotor layer (27 m $< z <$ 153 m). We also average the flow horizontally in the cross-stream (i.e., $y-$direction) direction. We distinguish between the flow immediately upstream of each front-row turbine (i.e., intra-turbine) and the flow in between turbines (i.e., inter-turbine). Figure 7 illustrates the velocity deficit $\Delta\overline{U}$ at hub height, defined as the difference between the time-averaged velocity at each grid cell $\overline{U}$ and the time-averaged velocity at the inflow of the domain $\overline{U}_\infty$. The intra-turbine region is shown as the stippled area in Figure 7a,b. The inter-turbine region corresponds
to the area in between the turbines (Figure 7a).

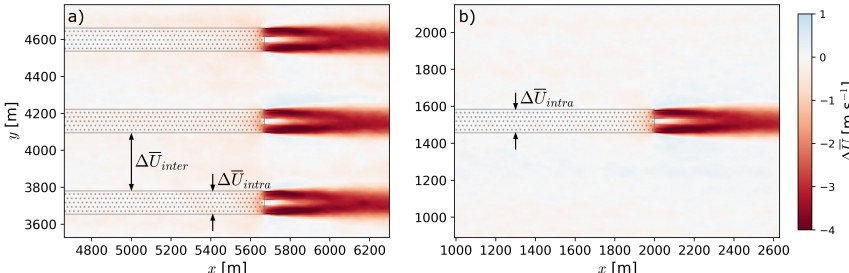

**Figure 7.** Horizontal velocity deficit ($\Delta\overline{U} = \overline{U} - \overline{U}_\infty$) at hub height for the wind plant (a) and stand-alone turbine (b) simulations. The velocity fields are averaged in time over 1 h of simulation. The stippled areas represent the intra-turbine regions. The inter-turbine regions are defined as the area in between turbines.

The velocity deceleration immediately upstream ($-2D < x < 0D$) of the wind plant is strongly influenced by the induction region of the individual turbines (Figure 8). The velocity deceleration in the inter- and intra-turbine regions differs substantially within $2D$ upstream of the first row of the wind plant, but is roughly equal farther than $2D$ upstream (Figure 8). The velocity deceleration asymptotes to zero far upstream ($x < -30D$) for both atmospheric conditions.

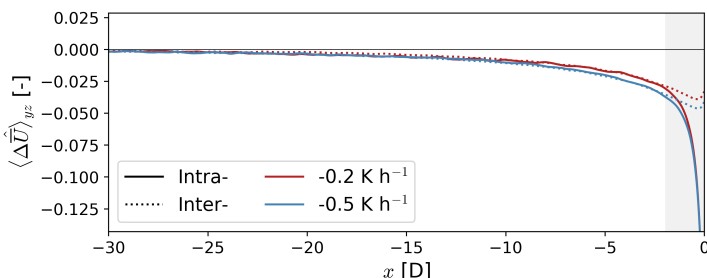

**Figure 8.** Normalized velocity deficit $\left(\Delta\hat{\overline{U}} = \frac{\overline{U} - \overline{U}_\infty}{\overline{U}_\infty}\right)$ for the inter- and intra-turbine regions upstream of the wind plant for each atmospheric condition. The velocity deficit is averaged vertically over the turbine rotor layer and horizontally (in the $y-$direction) over the corresponding area in Figure 7. The gray, shaded area represents the region over which the velocity deceleration is highly dependent on the individual turbines of the wind plant. The $x-$axis is scaled to locate $x = 0D$ at the location of the turbine.

The flow decelerates more upstream of a wind plant compared to a stand-alone turbine (Figure 9). To compare the velocity deficit between the wind plant and stand-alone turbine directly, we consider the velocity in the intra-turbine region only. On average for both atmospheric conditions, the velocity deceleration $2D$ upstream of the turbines is $1.57\%$ and $3.27\%$ for the stand-alone and wind plant, respectively. Likewise, the velocity deceleration extends farther upstream for a turbine array compared to a stand-alone turbine. Whereas the wind slows down by $1\%$ relative to the domain inflow at $3D$ upstream of a 190   stand-alone turbine, the same slowdown occurs $8.8D$ upstream of the wind plant.

     Wind plant blockage is amplified with atmospheric stability (Figure 9). The flow upstream of the first turbine row decelerates more in the moderate stability case compared to the weak stability case (solid lines in Figure 9). The horizontal wind speed deficit is on average $31\%$ slower for the moderate stability case between $x = -6D$ and $x = -2D$. For a stand-alone turbine, however, blockage does not change significantly for the stability regimes tested here (dashed lines in Figure 9).

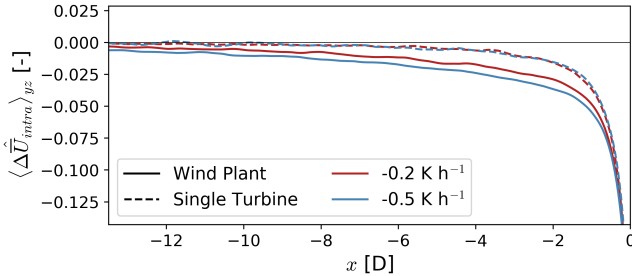

**Figure 9.** Normalized velocity deficit $\left(\Delta\hat{\overline{U}} = \frac{\overline{U} - \overline{U}_\infty}{\overline{U}_\infty}\right)$ upstream of the wind plant and stand-alone turbine for each atmospheric condition. The velocity deficit is averaged vertically over the turbine rotor layer and horizontally over the intra-turbine region (stippled area in Figure 7). The $x-$axis is scaled to locate $x = 0D$ at the location of the turbine.

Even though the wind speed slowdown from blockage is small, front-row turbines in the wind plant produce on average $5.2\%$ less power than a stand-alone turbine (Figure 10). Because winds are slightly faster in the moderate stability case compared

to the weak stability case, turbine power is also expected to differ. As a result, we evaluate the difference in power production between the turbines in the wind plant and a stand-alone turbine for the same atmospheric conditions. Just as the velocity deceleration is modified with atmospheric stability, turbine underperformance is more severe in the moderate stability case
compared to the weak stability case. Whereas turbines in the first row produce on average 4% less power than a stand-alone turbine for the weak stability condition, front-row turbines produce on average 6.5% less power than a stand-alone in the moderate stability case. Downstream of the first row of the wind plant, turbine power is primarily dominated by the evolution of the wake. Turbine wakes persist longer in stable boundary layers because of reduced turbulence mixing (e.g., Dörenkämper et al., 2015; Lee and Lundquist, 2017), so we expect downstream turbines to produce less power in the moderate stability case
compared to the weak stability case.

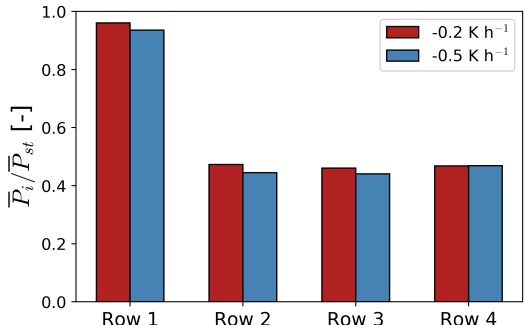

**Figure 10.** Normalized turbine power for each row of the wind plant and each atmospheric condition. The mean turbine power for the $i-$th row of the wind plant $\overline{P}_i$ is normalized over the mean turbine power of a stand-alone turbine $\overline{P}_{st}$.

## 4    Physical mechanisms modifying blockage

Differences in upstream blockage for both stability cases can be explained by evaluating the forcing mechanisms driving the flow. The steady state integral momentum equation for the $u-$velocity (Eq. 1) provides insight into the physical mechanisms acting on the flow. In Eq. 1, vector quantities are in bold, $\hat{\imath}$ is the unit vector in the $x-$direction, and $\hat{n}$ is the outward pointing
unit normal at each point on the surface $\mathcal{S}$ of the control volume $\mathcal{V}$.

$$\underbrace{\oint \rho \overline{u}\,(\overline{\boldsymbol{u}} \cdot \hat{\boldsymbol{n}})\,d\mathcal{S}}_{\text{momentum advection}} = \underbrace{\oint -\overline{p}\,(\hat{\boldsymbol{\imath}} \cdot \hat{\boldsymbol{n}})\,d\mathcal{S}}_{\text{pressure gradient}} + \underbrace{\int \rho f_c\,\overline{v}\,d\mathcal{V}}_{\text{Coriolis forcing}} - \underbrace{\int \nabla \cdot (\rho\,\overline{u'\boldsymbol{u}'})\,d\mathcal{V}}_{\text{turbulence mixing}} \tag{1}$$

We evaluate the balance between momentum advection by the mean flow and a pressure gradient. Even though the Coriolis force in our simulation domain is not negligible, Coriolis forcing in the induction region is small. The Coriolis parameter scales as $f_c \sim 10^{-4}$ s$^{-1}$ and the $v-$velocity in the turbine rotor layer for both stability cases is on the order of $\overline{v} \sim 0.1\,\mathrm{m\,s^{-1}}$, thus
Coriolis forcing is of the order $f_c\overline{v} \sim 10^{-5}$ m s$^{-2}$. Turbulence momentum redistribution is also small in the induction region of the wind plant for our simulations $\nabla \cdot (\overline{u'\boldsymbol{u}'}) \sim 10^{-4}$ m s$^{-2}$. In comparison, momentum advection by the mean flow in the induction region is of the order $10^{-1}$ m s$^{-2}$ in our simulations.

We evaluate the integral momentum equation on differential control volumes to examine the streamwise evolution of the flow (Figure 11). Each differential control volume $\delta\mathcal{V}$ (blue rectangular cuboid in Figure 11) is bounded vertically within the turbine rotor layer, and horizontally in the $y-$direction by the area covered by the wind plant. Along the $x-$direction, each differential control volume is $15\,\mathrm{m}$ long. For stand-alone turbine and a single front-row turbine in the wind plant, the control volume is bounded in the $y-$direction by the rotor diameter (Figure 11c,d). Because grid spacing is different for the stability cases, we interpolate atmospheric variables from each simulation to a common grid with horizontal resolution of 15 m.

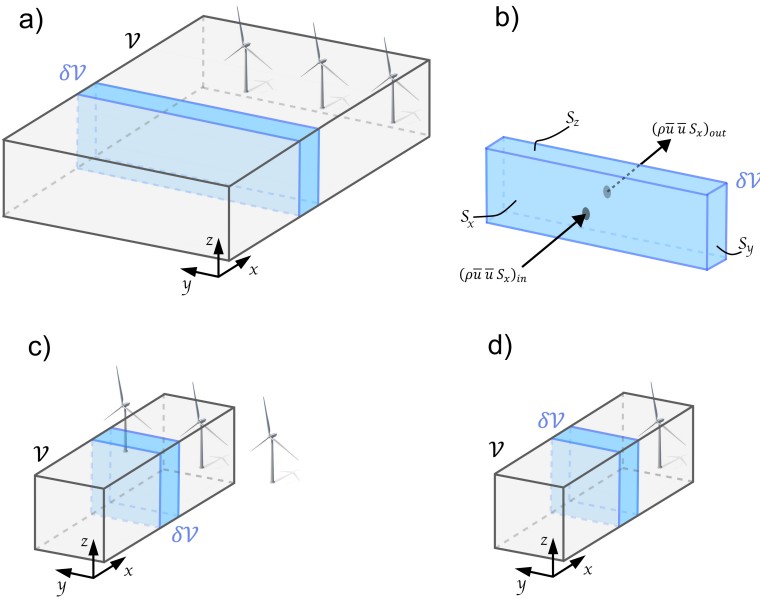

**Figure 11.** Illustration of the region considered in the analysis of the momentum balance along the $x-$direction for the whole wind plant (a), a single turbine in the front row of the wind plant (c) and stand-alone turbine (d). The integral momentum equation is evaluated on differential control volumes $\delta\mathcal{V}$ along the streamwise direction upstream of the turbines (b). Each control volume is bounded vertically by the top ($z=153\,\mathrm{m}$) and bottom ($z=27\,\mathrm{m}$) of the turbine rotor layer. Horizontally in the $y-$direction, the control volume spans the region upstream of the wind plant (from $y=1953\,\mathrm{m}$ to $y=5922\,\mathrm{m}$). For the single and stand-alone turbine (c,d), the control volume is bounded in the $y-$direction by the rotor diameter. Each differential control volume is $15\,\mathrm{m}$ long in the $x-$direction. The area of each control surface $S_i$ is illustrated in the differential control volume $\delta\mathcal{V}$ in Panel (b).

The balance between momentum advection and a pressure gradient along the $x-$direction for each differential control volume becomes:

$$\Delta(\rho\,\overline{u}\,\overline{u}\,S_x) + \Delta(\rho\,\overline{u}\,\overline{v}\,S_y) + \Delta(\rho\,\overline{u}\,\overline{w}\,S_z) = -\Delta\overline{p}S_x. \tag{2}$$

Each $\Delta(\rho\overline{u}\,\overline{u}_i\,S_i)$ term represents the net advection of $u-$momentum by the $u_i$ velocity component through the $S_i$ control surface. For example, the net advection of $u-$momentum by the $u-$velocity in a control volume $\delta\mathcal{V}$ is $\Delta(\rho\overline{u}\,\overline{u}\,S_x) = (\rho\overline{u}\,\overline{u}\,S_x)_{out} - (\rho\overline{u}\,\overline{u}\,S_x)_{in}$, as shown in Figure 11b.

The $u-$momentum flux at the inflow of the control volume $\mathcal{V}$ in Figure 11 is larger in the moderate stability case compared to the weak stability case due to slightly faster hub-height winds. Consequently, the magnitude of the momentum fluxes and turbine power is expected to be larger in the moderate stability case as well. To contrast the momentum balance between different atmospheric stability conditions and turbine array sizes, we normalize the forcing terms in Eq. 2 using the momentum flux at the inflow of the control volume far upstream $(\rho\overline{u}_\infty\overline{u}_\infty S_x)$ for each stability case.

The thrust force imparted by the turbine to the flow is a fundamental driver for blockage (Ebenhoch et al., 2017). In the numerical implementation of the GAD model, the aerodynamic forces are spread across multiple grid cells along the streamwise direction to avoid numerical instabilities (Mirocha et al., 2014). A pressure gradient forms in response to the thrust force that the turbine imparts on the flow ($\Delta\overline{p}_{pert}/\Delta x > 0$ upstream of the turbine in Figure 12). Because the thrust force is spread across multiple grid cells in the streamwise direction, the maximum in pressure in front of the turbines is located slightly upstream

of the actual location of the GAD in the numerical domain (Figure 12). As a result, we restrict the control volume $\mathcal{V}$ in Figure 11 to extend up to $x = 5647\,\mathrm{m}$, the location of the maximum in pressure perturbation upstream of the turbine array (vertical dotted line in Figure 12).

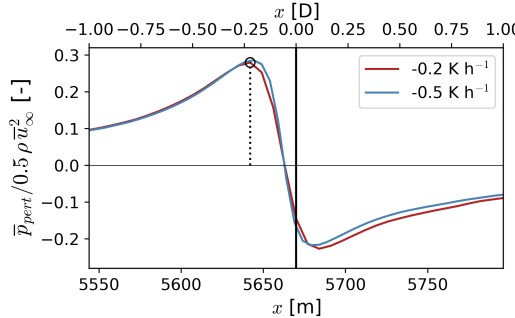

**Figure 12.** Hub-height pressure perturbation of a front-row turbine in the wind plant for each stability case. The pressure perturbation is normalized over the corresponding dynamic pressure for each stability condition. The solid black vertical line illustrates the location of the GAD in the numerical domain. The dotted vertical line illustrates the local maximum in pressure perturbation upstream of a front-row turbine in the wind plant. The secondary $x-$axis is scaled to locate $x = 0D$ at the location of the front-row turbine.

## 4.1  Stand-alone turbine

Flow deceleration upstream of a stand-alone turbine is primarily caused by an adverse pressure gradient (Figure 13). The

pressure gradient $\Delta\overline{p}S_x$ upstream of the turbine forms in response to the thrust force that the turbine imparts on the flow. Immediately upstream of the turbine (cross-hatched area in Figure 13), the pressure gradient force becomes negative because the GAD produces a pressure drop in the flow and the pressure perturbation field reaches a local maximum slightly upstream of

the turbine (Figure 12). In the numerical implementation of the GAD model, the aerodynamic forces are spread across multiple grid cells to avoid numerical instabilities (Mirocha et al., 2014), which causes the pressure field to decrease over multiple

grid cells (Figure 12). Momentum advection by the cross-stream $\Delta(\rho\overline{u}\overline{v}S_y)$ and vertical $\Delta(\rho\overline{u}\overline{w}S_z)$ velocity components also decrease momentum availability in the induction region of the turbine. Whereas the $v-$velocity transports momentum to both sides of the turbine, the $w-$velocity primarily transports momentum upwards. Immediately upstream of the turbines $(-1D < x < 0D)$, the vertical velocity is negative at the bottom of the turbine rotor layer (not shown), transporting momentum downwards. Nonetheless, the vertical velocity is positive over the rest of the induction region, transporting momentum upwards.

The streamwise velocity replenishes momentum in the induction region as the flow decelerates. Note that the momentum balance immediately upstream of the turbine (cross-hatched area in Figure 13) is not equal to zero because the thrust force from the GAD is not included in our calculations.

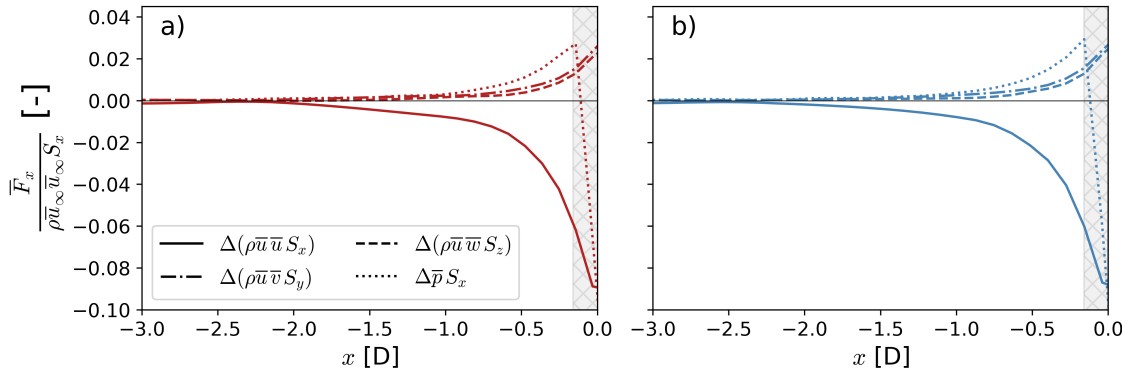

**Figure 13.** Streamwise evolution of the $u-$momentum equation (Eq. 2) for a stand-alone turbine in the weak (a) and moderate (b) stability cases. The integral momentum equation is evaluated on differential control volumes $\delta\mathcal{V}$ along the $x-$direction, as shown in Figure 11. The $x-$axis is scaled to locate $x = 0D$ at the location of the turbine. The mean momentum fluxes and the pressure gradient force are normalized using the $u-$momentum flux at the inflow of the control volume far upstream ($\rho\overline{u}_\infty\overline{u}_\infty S_x$) for the respective stability case. The cross-hatched area in each panel illustrates the grid cells influenced by the thrust force from the GAD.

The forcing mechanisms driving blockage for a stand-alone turbine remain virtually unchanged for the stability cases analyzed here (Figure 13). In the entire control volume $\mathcal{V}$ in Figure 11d, the pressure gradient force that drives flow deceleration

upstream of the turbine differs by 3.1% between atmospheric conditions. Similarly, $u-$momentum advection by the v- and $w-$velocity components varies by 3.7% and 1.8%, respectively, between atmospheric conditions.

### 4.2 Wind plant

Momentum balance in the streamwise direction indicates flow deceleration upstream of the first turbine row is balanced by a pressure gradient and vertical advection of horizontal momentum for both atmospheric conditions (Figure 14). An adverse

pressure gradient $\Delta\overline{p}S_x$ forms immediately upstream of each front-row turbine, producing a force that decelerates the flow. The pressure gradient force decays rapidly upstream of the turbines. Also depleting momentum within the turbine rotor layer,

the vertical advection of momentum $\Delta(\rho\overline{u}\,\overline{w}\,S_z)$ transports momentum upwards. Even though vertical advection of horizontal also decays upstream of the turbine array, the upwards transport of $u-$momentum remains larger than the pressure gradient force in the entire induction region of the wind plant for both atmospheric conditions. The cross-stream momentum advection $\Delta(\rho\overline{u}\,\overline{v}\,S_y)$ also reduces momentum availability, but only immediately upstream of the turbine array $(-0.5D < x < 0D)$. The streamwise velocity replenishes momentum in the region upstream of the first turbine row $\Delta(\rho\overline{u}\,\overline{u}\,S_x)$.

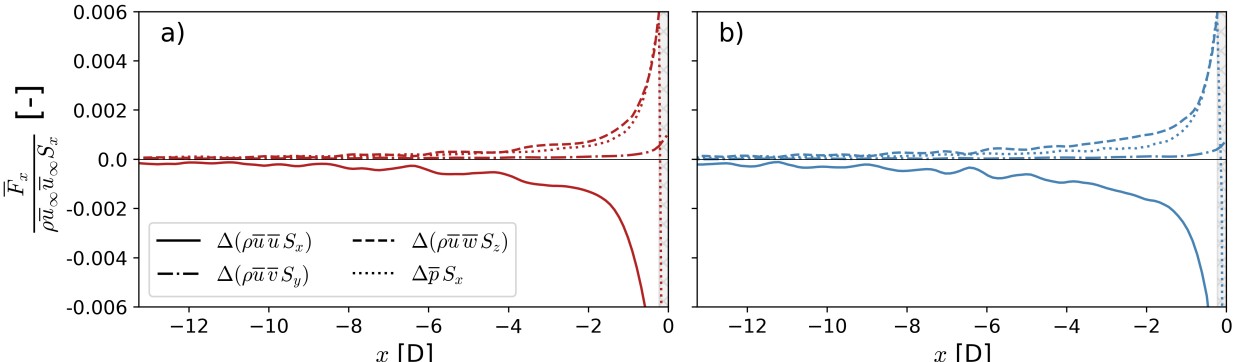

**Figure 14.** Streamwise evolution of the $u-$momentum equation (Eq. 2) for the weak (a) and moderate (b) stability cases. The integral momentum equation is evaluated on differential control volumes $\delta\mathcal{V}$ along the $x-$direction, as shown in Figure 11. The $x-$axis is scaled to locate $x = 0D$ at the location of the first turbine row. The mean momentum fluxes and the pressure gradient force are normalized using the momentum flux at the inflow of the control volume far upstream $(\rho\overline{u}_\infty\overline{u}_\infty\,S_x)$ for the respective stability case. The cross-hatched area in each panel illustrates the grid cells influenced by the thrust force from the GAD.

The vertical advection of streamwise momentum and adverse pressure gradient are the primary forcing mechanisms influencing wind plant blockage in our simulations. Figure 15 shows the net contribution of each term in Eq. 2 over the entire region upstream of the turbine array (control volume $\mathcal{V}$ in Figure 11a). Cumulatively over the induction region, vertical advection of horizonal momentum is 41.3% (18.4%) larger than the pressure gradient force for the moderate (weak) stability case. Momentum advection by the $v$-velocity is only 10.1% (12.8%) of the vertical advection of $u-$momentum for the moderate (weak) stability case. We now investigate how mean momentum transport and the adverse pressure gradient originate and compare within the induction region.

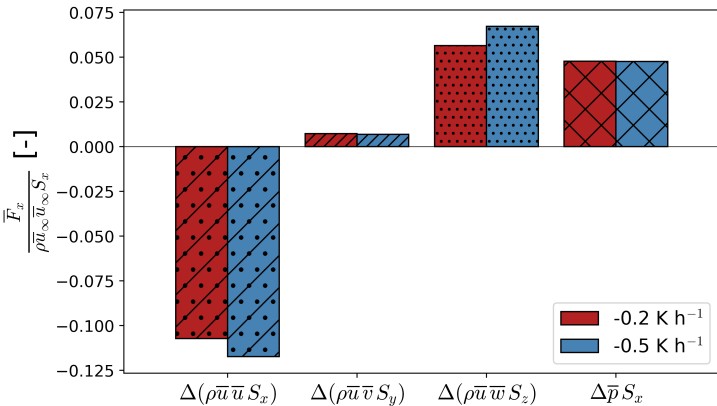

**Figure 15.** Momentum balance over the entire induction region of the wind plant. The integral momentum equation is evaluated on the control volume $\mathcal{V}$ shown in Figure 11a. The control volume $\mathcal{V}$ is bounded in the $x-$direction by the inflow of the domain and the maximum in pressure perturbation upstream of the turbines ($x = 5647\,\mathrm{m}$). The mean momentum fluxes and the pressure gradient force are normalized using the $u-$momentum flux at the inflow of the control volume far upstream ($\rho\,\overline{u}_\infty\overline{u}_\infty\,S_x$) for the respective stability case.

## 4.3 Pressure gradient force

Atmospheric stability marginally influences the streamwise pressure gradient upstream of the wind plant (Figure 16). The streamwise evolution of the normalized pressure gradient force remains nearly unchanged with atmospheric stability (Figure 16a). Furthermore, over the induction region, the normalized adverse pressure gradient upstream of the wind plant differs by 1% between the moderate and weak stability cases (Figure 16b).

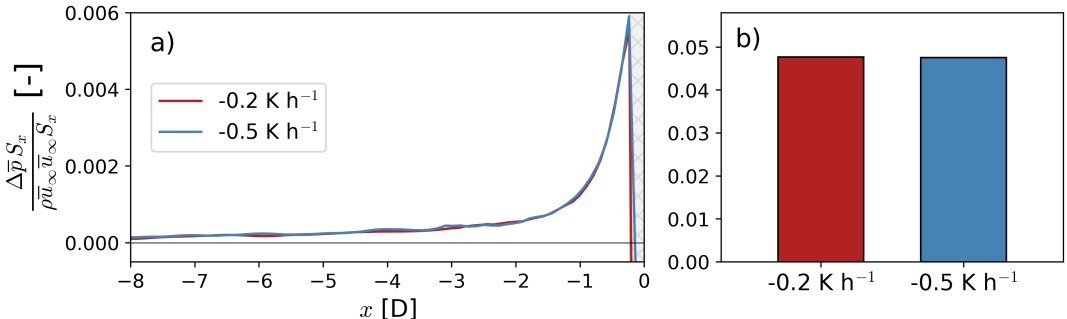

**Figure 16.** Streamwise evolution (a) and cumulative (b) pressure gradient force upstream of the wind plant. In panel (a), the integral momentum equation is evaluated on differential control volumes as shown in Figure 11a,b. In panel (b), the integral momentum equation is evaluated on the control volume $\mathcal{V}$ shown in Figure 11a. The pressure gradient force is normalized using the $u-$momentum flux at the inflow of the control volume far upstream ($\rho\,\overline{u}_\infty\overline{u}_\infty\,S_x$) for the respective stability case. The cross-hatched area in panel (a) illustrates the grid cells influenced by the thrust force from the GAD.

The pressure gradient upstream of a single front-row turbine in the wind plant is virtually the same as the pressure gradient force for a stand-alone turbine for both atmospheric conditions in our simulations (Figure 17). The streamwise evolution of the pressure gradient does not vary significantly between $-3D < x < 0D$ for a front-row turbine in the array and a stand-alone turbine (Figure 17a). Over the induction region of a stand-alone turbine $(-6D < x < 0D)$, differences in the pressure gradient force between a turbine in the wind plant and a stand-alone turbine are smaller than 3% (Figure 17b). Given that the normalized pressure gradient force remains unchanged with atmospheric stability and turbine array size, differences in blockage are caused by momentum redistribution in the induction region.

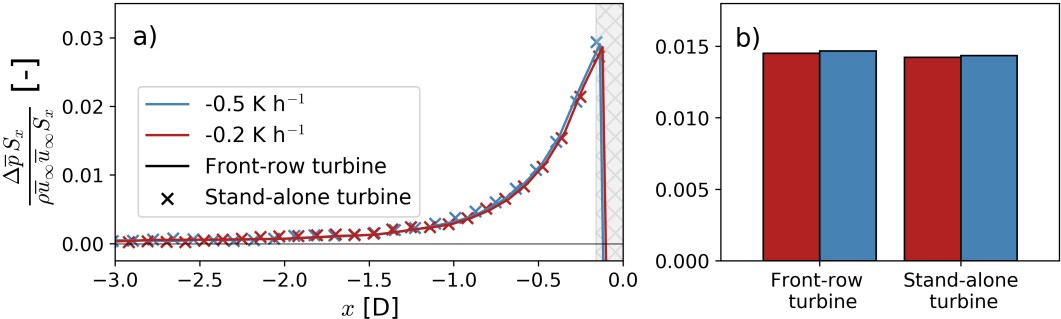

**Figure 17.** Streamwise evolution (a) and cumulative (b) pressure gradient force upstream of a single turbine for each stability case. In panel (a), the integral momentum equation is evaluated on differential control volumes as shown in Figure 11c for a single turbine in the middle of the wind plant and as shown in Figure 11d for a stand-alone turbine. In panel (b), the integral momentum equation is evaluated on the control volume $\mathcal{V}$ shown in Figures 11c,d for a single turbine in the middle of the wind plant and for a stand-alone turbine. The pressure gradient force is normalized using the $u-$momentum flux at the inflow of the control volume $\mathcal{V}$ in Figures 11c,d far upstream $(\rho \overline{u}_\infty \overline{u}_\infty S_x)$ for the respective stability case. The cross-hatched area in panel (a) illustrates the grid cells influenced by the thrust force from the GAD.

## 4.4 Mean momentum advection

The streamwise flow deceleration in the induction region is primarily transferred into upward motions (Figure 18). We evaluate the integral mass conservation equation $\oint \rho (\overline{u} \cdot \hat{n}) \, d\mathcal{S} = 0$ on the differential control volumes shown in Figure 11a,b. Mass balance indicates the slowdown of the $u-$velocity in the turbine rotor layer $(\Delta(\rho \overline{u} S_x) < 0)$ is balanced by the development of a secondary flow feature in the form of net-upwards vertical motion $(\Delta(\rho \overline{w} S_z) > 0)$ for both stability conditions (i.e., $\Delta(\rho \overline{u} S_x) + \Delta(\rho \overline{w} S_z) \approx 0$). The development of the vertical velocity is possible because of a vertical pressure gradient that balances the downward buoyancy force in the stably stratified flow (see Appendix B for a deeper analysis on vertical momentum balance). The change in $v-$velocity to the sides of the wind plant is only significant immediately in front of the turbine array $(-0.5D < x < 0D)$.

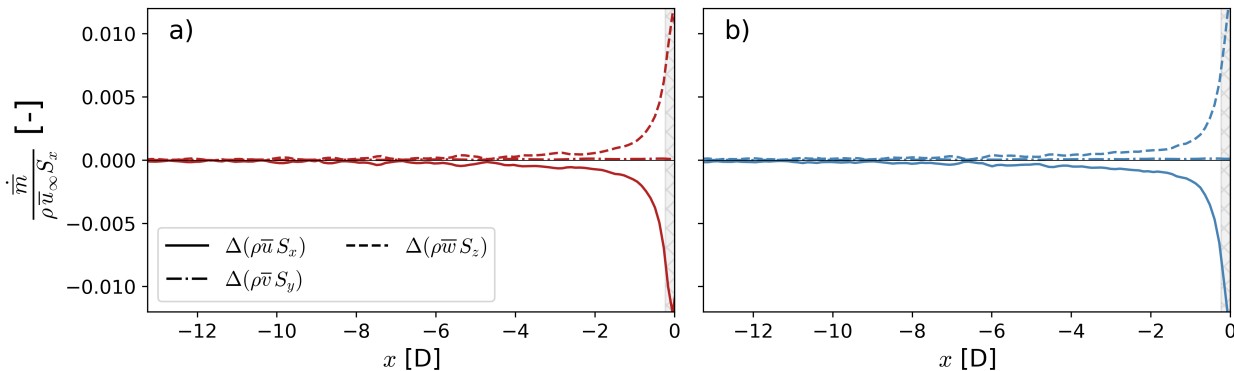

**Figure 18.** Streamwise evolution of the mass conservation equation for the weak (a) and moderate (b) stability cases. The integral mass conservation equation is evaluated on differential control volumes $\delta\mathcal{V}$ along the $x-$direction, as shown in Figure 11. The $x-$axis is scaled to locate $x = 0D$ at the location of the first turbine row. The mass fluxes are normalized using the mass flux at the inflow of the control volume far upstream ($\rho\,\overline{u}_{\infty}\,S_x$) for the respective stability case. The cross-hatched area in each panel illustrates the grid cells influenced by the thrust force from the GAD.

The vertical velocity advects horizontal momentum out of the turbine rotor layer (Figure 19). Vertical advection of horizontal momentum is 20% larger in the moderate stability case compared to the weak stability upstream of the first turbine row (Figure 19b). Larger vertical shear of the horizontal velocity in the moderate stability case compared to the weak stability case is the primary cause for the increased vertical advection of horizontal momentum. Shear $\left(\frac{\Delta\overline{u}}{\Delta z} = \frac{\overline{u}_t - \overline{u}_b}{D}\right)$ between the bottom $\overline{u}_b$ and top $\overline{u}_t$ of the turbine rotor layer is 43.6% larger in the $-0.5\,\mathrm{K\,h}^{-1}$ simulation compared to the $-0.2\,\mathrm{K\,h}^{-1}$ simulation. Similarly, the vertical velocity in the turbine rotor layer is 20% larger in the moderate stability case than in the weak stability case between $x = -6D$ and $x = 0D$. The vertical velocity is expected to be larger in the moderate stability case because, as shown in Figure 18, the streamwise slowdown of the flow is transformed almost entirely into vertical motions. As a result, advection of horizontal momentum by the vertical velocity $\Delta(\rho\,\overline{u}\,\overline{w}\,S_z) = \rho\,S_z(\overline{u}_t\,\overline{w}_t - \overline{u}_b\,\overline{w}_b)$ is amplified.

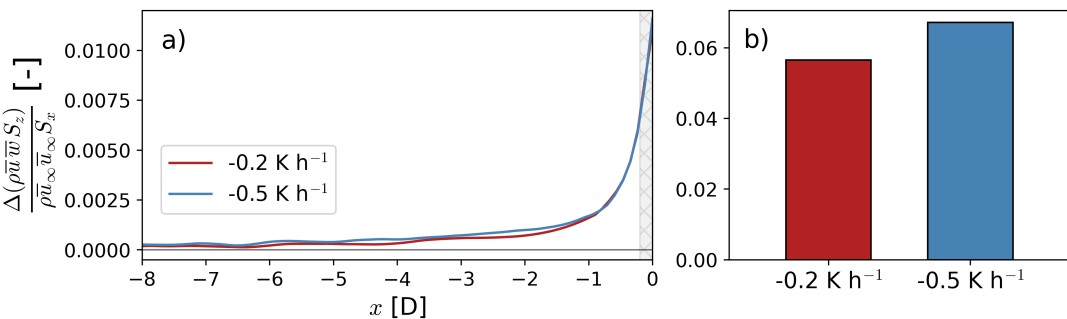

**Figure 19.** Streamwise evolution (a) and cumulative (b) vertical advection of $u-$momentum upstream of the wind plant. In panel (a), the integral momentum equation is evaluated on differential control volumes as shown in Figure 11a,b. In panel (b), the integral momentum equation is evaluated on the control volume $\mathcal{V}$ shown in Figure 11a. The vertical momentum flux is normalized using the $u-$momentum flux at the inflow of the control volume far upstream $(\rho\overline{u}_\infty\overline{u}_\infty S_x)$ for the respective stability case. The cross-hatched area in panel (a) illustrates the grid cells influenced by the thrust force from the GAD.

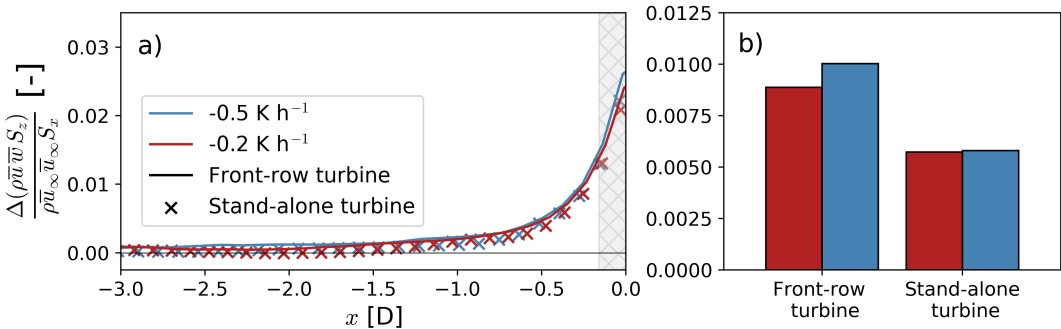

**Figure 20.** Streamwise evolution (a) and cumulative (b) vertical advection of $u-$momentum upstream of a single turbine for each stability case. In panel (a), the integral momentum equation is evaluated on differential control volumes as shown in Figure 11c for a single turbine in the middle of the wind plant and as shown in Figure 11d for a stand-alone turbine. In panel (b), the integral momentum equation is evaluated on the control volume $\mathcal{V}$ shown in Figures 11c,d for a single turbine in the middle of the wind plant and for a stand-alone turbine. The pressure gradient force is normalized using the $u-$momentum flux at the inflow of the control volume $\mathcal{V}$ in Figures 11c,d far upstream $(\rho\overline{u}_\infty\overline{u}_\infty S_x)$ for the respective stability case. The cross-hatched area in panel (a) illustrates the grid cells influenced by the thrust force from the GAD.

Vertical advection of horizontal momentum is amplified for a wind plant compared to a stand-alone turbine (Figure 20). For a given atmospheric condition, vertical shear of the horizontal velocity remains unchanged between the stand-alone turbine and wind plant simulations. Therefore, differences in vertical transport of horizontal momentum between a stand-alone turbine and a turbine in the wind plant are entirely due to the vertical velocity that forms upstream of the turbine array (Figure 21). For the wind plant, the secondary flow (i.e., net upwards $w-$velocity) extends farther upstream than for a stand-alone turbine (Figure 21). Whereas vertical advection of horizontal momentum is on average 26% larger for a front-row turbine in the array compared

to stand-alone turbine between $-1D < x < 0D$, it is 2 times larger between $-6D < x < -1D$. Over the induction region of a stand-alone turbine ($-6D < x < 0D$), vertical transport of horizontal momentum is 72% (55%) larger for a front-row turbine in the array than for a stand-alone turbine for the moderate (weak) stability case (Figure 20b).

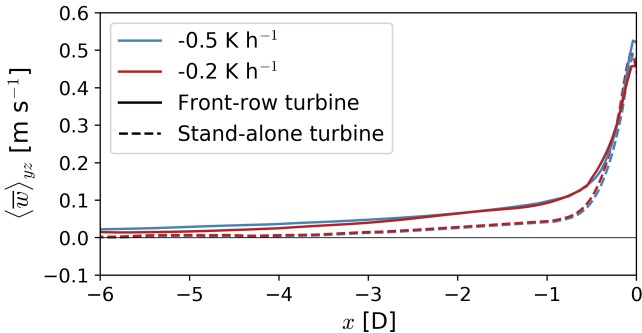

**Figure 21.** Streamwise evolution of the vertical velocity upstream of a stand-alone and front-row turbine in middle of the wind plant. The vertical velocity is averaged in the $y-$direction over the rotor diameter and in the $z-$direction over the top half of the rotor layer.

## 5   Discussion and conclusions

The horizontal wind component within the rotor swept area decelerates upstream of wind plants, deflecting and transporting
momentum upward, a phenomenon called blockage. Blockage undermines energy production of wind plants by reducing turbine power production of mainly the front-row turbines. As the magnitude of the velocity deceleration and the nature of the physical mechanisms amplifying blockage are not yet well understood, we perform idealized WRF-LES of a finite-size wind plant and a stand-alone turbine for two atmospheric conditions to characterize blockage in stable boundary layers. Furthermore, we analyze the mechanisms driving and amplifying blockage by evaluating momentum conservation upstream of the turbines.
The velocity deceleration upstream of a wind plant is larger than upstream of a stand-alone turbine. For our simulations, the velocity deficit due to blockage is twice as large for the wind plant compared to a stand-alone turbine 2D upstream (Figure 9). Likewise, Reynolds-averaged Navier-Stokes simulations (RANS) by Bleeg et al. (2018) show the velocity slowdown 2D upstream of a wind plant is on average 1.9 times larger than for a stand-alone turbine. For a variety of wind plant layouts, Strickland and Stevens (2022) and Strickland et al. (2022) also show the velocity deceleration 2D upstream is consistently
larger for a turbine array compared to a single turbine in isolation. The velocity deficit in the induction region also extends farther upstream of a wind plant compared to a single turbine. For the simulations herein, the velocity deficit 7D upstream of a stand-alone turbine is negligible, whereas the deficit at the same distance for a wind plant is on average 1.2%. Comparably, Bleeg et al. (2018) suggest isolated turbines do not influence the flow $7-10D$ upstream, but wind plants do. However, it should be noted that Bleeg et al. (2018) do not provide a specific description of the stability cases or of the size of the wind plants used
in their simulations, preventing a direct comparison.

The dominant physical mechanism that decelerates the flow upstream of a wind plant is different than for a stand-alone turbine in our simulations. Blockage for a stand-alone turbine is primarily caused by a pressure gradient upstream (Figure 13). The pressure gradient force is also present upstream of the wind plant; however, the vertical advection of horizontal momentum contributes more to the deceleration of the flow (Figure 15). The pressure gradient force upstream of a front-row turbine of the wind plant is practically the same as for a stand-alone turbine (Figure 17). Conversely, the vertical transport of $u-$momentum upstream of a front-row turbine of the wind plant is on average 63% larger than for a stand-alone turbine between $-6D < x < 0D$ (Figure 20).

Vertical advection of $u-$momentum is larger for a wind plant compared to a stand-alone turbine because of a secondary flow feature that forms upstream of the turbine array. The slowdown of the $u-$velocity in the induction region of the wind plant is transferred into vertical motions (Figure 18). Other simulation studies have also noted this vertical deflection of the flow (e.g., Wu and Porté-Agel, 2017; Allaerts and Meyers, 2017). The vertical velocity advects horizontal momentum out of the turbine rotor layer. Given that the secondary flow extends far upstream of the turbine array (Figure 21), vertical advection of horizontal momentum is larger for the wind plant compared to the stand-alone turbine (Figure 20).

Boundary layer stability amplifies blockage for a wind plant (Figure 9). The wind speed is 3.5% and 2.8% slower than freestream 2D upstream of the wind plant for moderately and weakly stably stratified flow, respectively (solid lines in Figure 9). For a stand-alone turbine, atmospheric stability has only a second order effect on blockage. The wind slows down by 1.55% and 1.60% 2D upstream of a stand-alone turbine for the weak and moderate stability cases, respectively (dashed lines in Figure 9). Bleeg and Montavon (2022) and Strickland et al. (2022) also found wind plant blockage changes with stability. Bleeg and Montavon (2022) quantify stability using the vertical change in potential temperature in the boundary layer $\Delta\theta_{bl}$. For a 600 m deep boundary layer with weak temperature stratification ($\Delta\theta_{bl} = 0.6\,\mathrm{K}$), they report a single turbine row produces on average 2.2% less power than a turbine in isolation (Bleeg and Montavon, 2022). For a $600\,\mathrm{m}$ deep boundary layer with moderate stratification ($\Delta\theta_{bl} = 2.9\,\mathrm{K}$), they report a single turbine row produces on average 7.2% less power than a turbine in isolation (Bleeg and Montavon, 2022). In comparison, front-row turbines in our weak ($z_{bl} = 660\,\mathrm{m}$, $\Delta\theta_{bl} = 1.37\,\mathrm{K}$) and moderate ($z_{bl} = 360\,\mathrm{m}$, $\Delta\theta_{bl} = 2.63\,\mathrm{K}$) stability simulations produce 4% and 6.5% less power than a stand-alone turbine, respectively (Figure 10). The differences in our results can arise from very different wind plant layouts, as blockage has been shown to be sensitive to turbine spacing and wind plant size (Strickland and Stevens, 2022; Bleeg and Montavon, 2022). Bleeg and Montavon (2022) perform RANS of 21 turbines in a single turbine row, while we do LES of a four-row wind plant with 10 turbines per row. The fact that the velocity slowdown herein remains unchanged with atmospheric stability for a stand-alone turbine but not for a wind plant suggests differences in blockage for the wind plant are due to the large-scale interaction between the turbine array and the boundary layer.

The larger velocity deceleration upstream of a wind plant in the moderate stability case compared to the weak stability case is due to increased vertical transport of horizontal momentum upstream of the first turbine row (Figure 15). The normalized pressure gradient in the induction region remains unchanged with atmospheric stability for the wind plant as a whole (Figure 16) and for individual turbines in the array (Figure 17). Conversely, vertical advection of $u-$momentum upstream of the wind plant as a whole (Figure 19) and for individual turbines in the array (Figure 20) is larger in the moderate stability case compared

to the weak stability case. Vertical shear of the horizontal velocity is 43% larger in the moderate stability case compared to the weak stability case, contributing to increased vertical transport of horizontal momentum in the induction region. Bleeg and Montavon (2022) also highlight the importance of vertical shear of the horizontal velocity and the vertical deflection of the flow. They suggest that, due to shear, the hub-height flow at the turbine location is slower than far upstream because the flow is being deflected upwards (Bleeg and Montavon, 2022). Potential flow models often used to include the power loss from blockage in energy assessments do not account for vertical shear of the horizontal velocity (e.g., Forsting et al., 2021). Vertical advection of $u-$momentum is the primary amplifier for blockage in our simulations, driven by shear. As a result, energy yield estimates might underestimate losses from blockage.

Other studies analyzing the physical mechanism modifying blockage (with minimal upstream propagation of gravity waves) conclude the adverse pressure gradient amplifies blockage with closely spaced turbines (Strickland and Stevens, 2022; Strickland et al., 2022). Even though Strickland et al. (2022) demonstrate kinetic energy fluxes are the dominant mechanism modifying the power production of front-row turbines, they suggest an increased pressure gradient amplifies blockage as cold air is deflected upwards (i.e., $u-$velocity is transformed into $w-$velocity). Note that we quantify the pressure gradient force upstream of the turbine array, which includes contributions from horizontal density gradients (i.e., cold air anomaly), and do not find significant changes with atmospheric stability (Figure 16). Strickland and Stevens (2022) and Strickland et al. (2022) do not consider the influence of momentum advection upstream of the turbines; nonetheless, they show increasingly larger vertical velocity upstream of the turbines with smaller cross-stream turbine spacing and stronger atmospheric stability, likely increasing momentum advection outward of the turbine rotor layer. Furthermore, Strickland and Stevens (2022) simulate purely neutral flow whereas we simulate stably stratified flow in a boundary layer with a capping inversion above. Strickland et al. (2022) simulate an infinitely wide wind plant without a capping inversion. Bleeg and Montavon (2022) show the velocity reductions from blockage, and thus the amplifying mechanisms, vary when a capping inversion, a typical feature of the planetary boundary layer, is not included in the simulations.

Thus, while wind plant blockage is caused by an adverse pressure gradient forming upstream the turbines, it is amplified by the vertical advection of horizontal momentum in the turbine rotor layer for the stability cases analyzed here. A pressure gradient forms from the thrust force of the individual turbines. The cumulative deceleration of the flow upstream of the individual turbines is balanced by an increase in vertical velocity. Vertical motions transport horizontal momentum out of the turbine rotor layer, reducing momentum availability at the turbine locations.

It is important to highlight that our simulations are idealized and in flat terrain. The role of terrain and vegetation should also be considered when evaluating wind plant blockage in future studies. Not only does terrain complicate efforts to measure blockage (Sanchez Gomez et al., 2022), but it likely also modifies the momentum balance upstream of the turbines (Segalini, 2017). Another important area of future research is whether or not wind plants trigger gravity waves, which may amplify blockage. Future field experiments should evaluate gravity-wave initiation around wind plants. Knowledge about interactions between gravity waves and wind plants may provide a better understanding of the physics of wind plant blockage and aid in model validation. In addition, more research is needed to further validate the forcing mechanisms driving blockage for a front-row turbine in the wind plant and a stand-alone turbine for a wide range of atmospheric conditions.

*Code availability.* The WRF model v4.1.5 used herein is available at: https://github.com/miguel-sg-2/WRF_versions.git. The namelist.input files for each simulation and the turbine specifications are available for download at DOI:10.5281/zenodo.7604167.

## Appendix A: Tiling approach

Simulating stable boundary layers using nested LES for large domains ($L_x, L_y \sim 10^4$ m) requires sizable computational re-
410 sources due to a long spin-up time, and small time steps associated with the fine horizontal and vertical-grid required to resolve turbulence. Most studies using a nested approach run their full-size domain for $\sim 10$ h to obtain a fully turbulent, stable boundary layer (e.g., Kosović and Curry, 2000; Muñoz-Esparza and Kosović, 2018; Peña et al., 2021). In such a way, we would run a $\sim 10$ h long simulation with $345 \times 10^6$ ($112 \times 10^6$) grid points for the moderate (weak) stability case to obtain a fully developed stable boundary layer. We develop an alternative approach that reduces computational requirements during the spin-up
phase of the simulation by a factor of 25. We spin-up a fully developed stable boundary layer in a small ($L_x, L_y \sim 10^3$ m) precursor domain. After the flow is fully turbulent and stably stratified, we tile multiple precursor domains along the $x-$ and $y-$directions to form a large domain. Given that the large domain is composed of an array of identical smaller domains, we let turbulence break periodicity in the flow before initializing the nested domain, where the turbines will interact with the flow.

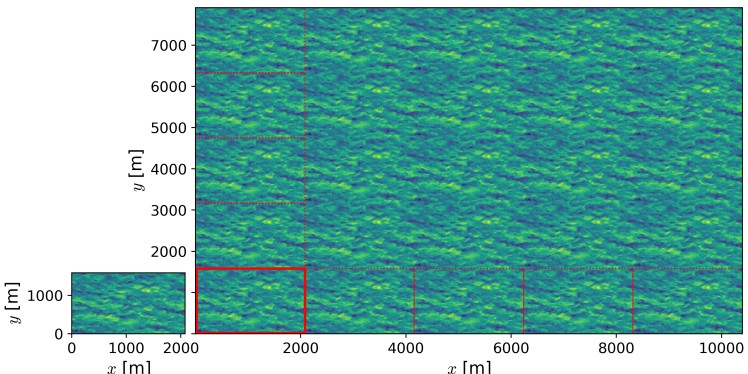

**Figure A1.** Plan view of hub-height wind speed for the precursor domain after spin-up, and for the large LES domain at initialization. Multiple precursor domains are tiled along the horizontal directions (5 along each direction) to form a large domain. The red rectangles illustrate the relative size and location of the tiles used to form the large domain.

A large, periodic LES domain is initialized by tiling small, fully turbulent domains along the horizontal directions. Figure
A1 exemplifies the tiling procedure for one of our LES domains at initialization. Boundary conditions for the resulting large LES domain are satisfied because of the periodic nature of the precursor LES domains.

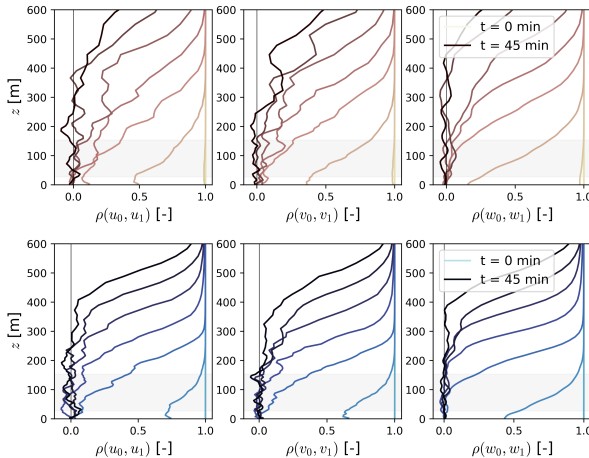

**Figure A2.** Correlation of atmospheric variables between adjacent tiles in the large LES domain for the weak (top) and moderate (bottom) stability cases. Profiles are color-coded in $5\,\mathrm{min}$ increments since initialization of the large domain. The grey, shaded region represents the turbine rotor layer.

Flow at initialization displays periodicity along the horizontal directions due to the tiling procedure (Figure A2). As expected, atmospheric variables between adjacent tiles are perfectly correlated ($\rho = 1$) at initialization. We let the flow break periodicity before introducing the turbines in the simulation. The velocity field becomes uncorrelated from the bottom upwards. Turbulence production close to the surface breaks periodicity in the flow, which then propagates upwards (Figure A2). It takes $40\,\mathrm{min}$ ($35\,\mathrm{min}$) for the flow in the boundary layer to be uncorrelated between adjacent tiles for the weak (moderate) stability case. We expect flow above the boundary layer to remain correlated between adjacent tiles because turbulence in that region is small. Figure A3 shows the flow at hub height 30 min after initialization, where turbulence structures are no longer correlated between adjacent tiles.

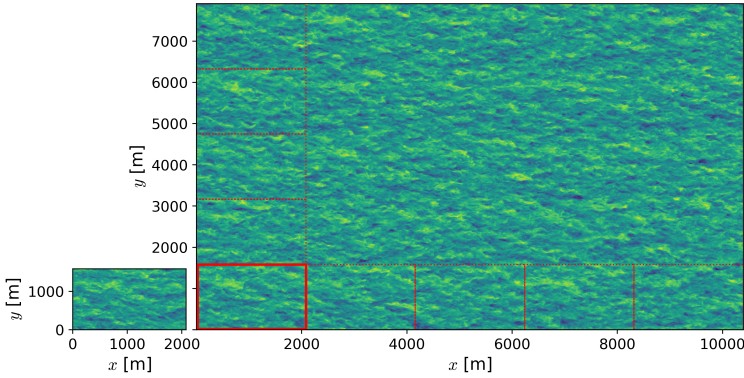

**Figure A3.** As in Figure A1, but after spin-up of the large domain is complete.

The mean flow and turbulence statistics in the large domain are comparable to that of the precursor domain (Figure A4). The Pearson correlation coefficient between the atmospheric variables ($\langle u \rangle_{xy}, \langle v \rangle_{xy}, \langle k \rangle_{xy}, \langle w'w' \rangle_{xy}$) of the precursor and large domain remains above 0.995 after the flow in the large domain is uncorrelated for adjacent tiles. The only significant difference between the precursor and large domain is in turbulence statistics close to the surface. Turbulence quantities are larger for the large domain, likely due to larger-scale turbulence structures being able to form within the larger domain.

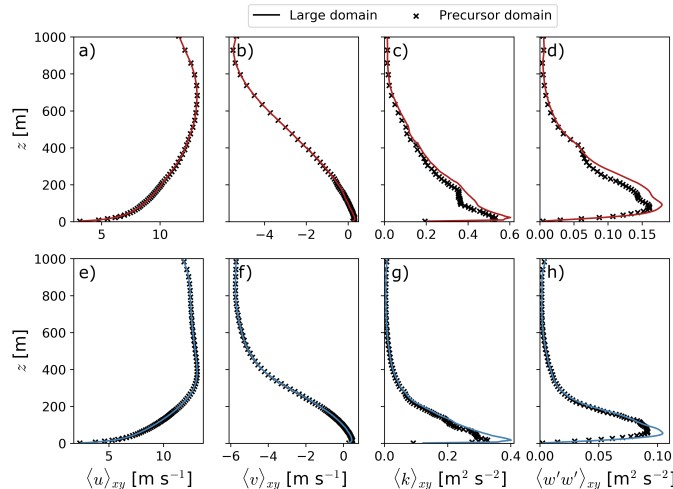

**Figure A4.** Mean flow and turbulence statistics for the precursor and large LES domains for the weak (top) and moderate (bottom) atmospheric stability cases.

A nested LES domain with turbines represented as actuator disks is initialized after boundary layer flow is no longer periodic.

## Appendix B: Vertical momentum balance

We evaluate the steady state integral momentum equation for the $w-$velocity (Eq. B1) on differential control volumes (Figure 11) to analyze the forcing for vertical flow. In Eq. B1, vector quantities are in bold, $\hat{k}$ is the unit vector in the $z-$direction, and $\hat{n}$ is the outward pointing unit normal at each point on the surface $S$ of the control volume $\mathcal{V}$.

$$\underbrace{\oint \rho \overline{w} (\overline{\boldsymbol{u}} \cdot \hat{\boldsymbol{n}}) \, dS}_{\text{momentum advection}} = \underbrace{\oint -\overline{p} (\hat{\boldsymbol{k}} \cdot \hat{\boldsymbol{n}}) \, dS}_{\text{pressure gradient}} - \underbrace{\int \rho g' \, d\mathcal{V}}_{\text{buoyancy}} \tag{B1}$$

The buoyancy force in Eq. B1 is given in terms of reduced gravity $g' = g \frac{\rho - \rho_\infty}{\rho_\infty}$, where $\rho_\infty$ is the fluid density at the inflow of the domain. The pressure gradient in Eq. B1 represents the deviation from hydrostatic balance.

The balance between momentum advection, a pressure gradient and buoyancy for each differential control volume becomes:

$$\Delta(\rho \overline{w} \, \overline{u} \, S_x) + \Delta(\rho \overline{w} \, \overline{v} \, S_y) + \Delta(\rho \overline{w} \, \overline{w} \, S_z) = -\Delta \overline{p} S_z - \rho g' V. \tag{B2}$$

Each $\Delta(\rho\overline{w}\,\overline{u}_i\,S_i)$ term represents the net advection of $w-$momentum by the $u_i$ velocity component through the $S_i$ control surface. For example, the net advection of $w-$momentum by the $u-$velocity in a control volume $\delta\mathcal{V}$ is $\Delta(\rho\overline{w}\,\overline{u}\,S_x) = (\rho\overline{w}\,\overline{u}\,S_x)_{out} - (\rho\overline{w}\,\overline{u}\,S_x)_{in}$. To contrast the momentum balance between different atmospheric stability conditions and turbine array sizes, we normalize the forcing terms in Eq. B2 using the $u-$momentum flux at the inflow of the control volume far upstream $(\rho\overline{u}_\infty\overline{u}_\infty\,S_x)$.

Figure B1 shows the streamwise balance of vertical momentum. Because of the convention adopted throughout the manuscript ($\Delta X = X_{out} - X_{in}$ for an arbitrary variable $X$ on the control volume $\mathcal{V}$ shown in Figure 11), negative terms in Figure B1 correspond to upward forces. As such, $\Delta\overline{p}S_z < 0$ is forcing the flow upwards and $\rho g'V > 0$ is forcing the flow downwards.

Momentum balance for the $w-$velocity indicates the secondary flow (i.e., $w-$velocity) in the induction region is driven by a pressure gradient far upstream, and horizontal transport of $w-$momentum close to the turbines (Figure B1). Immediately upstream of the first turbine row $(-1D < x < 0D)$, horizontal advection of $w-$momentum drives upward motions. The sharp deceleration of the $u-$velocity immediately upstream of each front-row turbine ($\Delta\overline{u} < 0$ is large as shown in Figure 9) results in momentum replenishment, which is balanced by a downward pressure gradient force ($\Delta\overline{p}S_z > 0$). Farther upstream ($x < -1D$), an upward pressure gradient force ($\Delta\overline{p}S_z < 0$) overcomes buoyancy and the streamwise advection of vertical momentum. Redistribution of vertical momentum by the $v-$ and $w-$velocity components is marginal within the induction region of the wind plant.

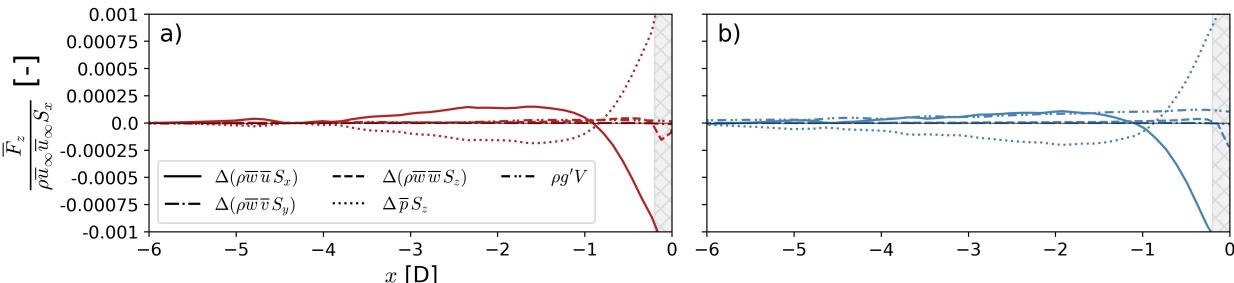

**Figure B1.** Streamwise evolution of the $w-$momentum equation (Eq. B1) for the weak (a) and moderate (b) stability cases. The integral momentum equation is evaluated on differential control volumes $\delta\mathcal{V}$ along the $x-$direction, as shown in Figure 11. The $x-$axis is scaled to locate $x = 0D$ at the location of the first turbine row. The mean momentum fluxes, the pressure gradient force and buoyancy are normalized using the $u-$momentum flux at the inflow of the control volume far upstream $(\rho\overline{u}_\infty\overline{u}_\infty\,S_x)$. The cross-hatched area in each panel illustrates the grid cells influenced by the thrust force from the GAD.

Even though the flow in our simulations is stably stratified, the effect of buoyancy is only significant in the moderate stability case (Figure B1). The vertical displacement of the flow in the induction regions is small, resulting in small changes in density over the induction region. For both stability cases, the streamline displacement at $z = 153\,\mathrm{m}$ between the inflow of the domain and the first turbine row is smaller than 10 m. Consequently, the fractional change in fluid density in the induction region is 0.004% and 0.0002% for the moderate and weak stability cases, respectively.

## Appendix C: Gravity waves

We examine the upstream propagation of gravity waves in our simulations by analyzing the correlation between the pressure, vertical velocity and potential temperature fields. We evaluate the streamwise deviation of each atmospheric variable from the inflow of the domain. Each variable is averaged over region upstream of the wind plant (from $y = 1953\,\mathrm{m}$ to $y = 5922\,\mathrm{m}$). Furthermore, we normalize each variable $a_i$ as $\hat{\hat{a}}_i = \frac{a_i}{max(a_i) - min(a_i)}$, so that its values are between -1 and 1. Figure C1 shows the streamwise evolution of the deviation in vertical velocity, pressure and potential temperature from the inflow of the domain at the capping inversion.

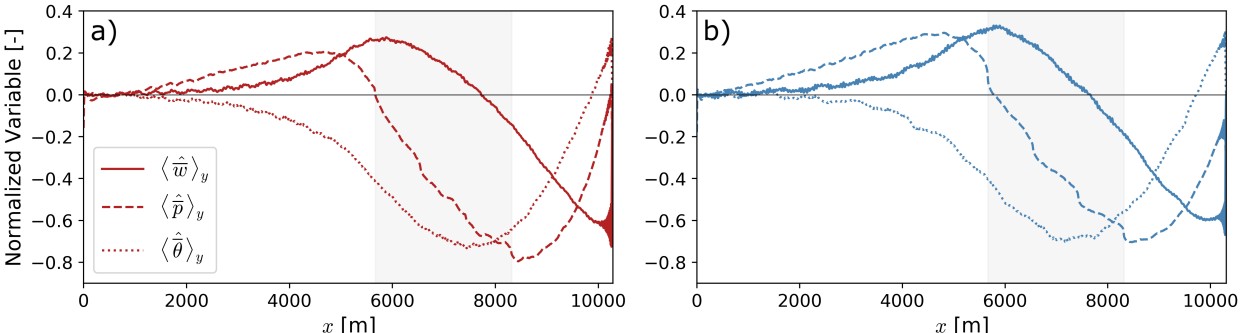

**Figure C1.** Streamwise evolution of the vertical velocity, pressure and potential temperature deviation from inflow conditions for the weak (a) and moderate (b) stability cases at $z = 1200\,\mathrm{m}$. Each variable $a_i$ is normalized as $\hat{\hat{a}}_i = \frac{a_i}{max(a_i) - min(a_i)}$ and averaged along the $y-$direction (from $y = 1953\,\mathrm{m}$ to $y = 5922\,\mathrm{m}$). The gray shaded area in each panel represents the region covered by the wind plant.

There is no evidence of wind plant-triggered upstream-propagating gravity waves in our simulation domain for either atmospheric condition. In our simulations, the pressure and vertical velocity are out of phase by $\sim 90°$ (Figure C1). Upstream of the wind plant, the vertical velocity perturbation reaches a maxima when the pressure perturbation is zero. At the outflow of the wind plant, the pressure perturbation is at a minima and the vertical velocity perturbation is close to zero. In internal gravity waves, the pressure perturbation is in phase with the vertical velocity perturbation (Banta et al., 1990). Furthermore, our simulations show the pressure perturbation and the potential temperature perturbation are out of phase by $\sim 45°$ (Figure C1). Conversely, the potential temperature perturbation and the pressure perturbation are $90°$ out of phase in gravity waves (Banta et al., 1990). Note that we show the phase shift between the pressure, vertical velocity and potential temperature above the capping inversion; however, this phase shift is also observed in the boundary layer.

Spurious waves can sometimes modify the correlation between atmospheric variables in upstream-propagating gravity waves (Lanzilao and Meyers, 2022). Because the only potential source of gravity waves in our simulations is in the boundary layer (i.e., the wind plant), then waves with a downward group velocity (positive phase speed) and outside the boundary layer must be due to spurious reflections (Taylor and Sarkar, 2007). We quantify wave reflection following the methodology outlined in Taylor and Sarkar (2007). We find that 7.1% and 5.8% of the total vertical kinetic energy $0.5w'^2$ is associated with downward

energy propagation for the weak and moderate stability cases, respectively, which is comparable to the wave reflection reported in other studies (Taylor and Sarkar, 2007; Allaerts and Meyers, 2017, 2018).

We also evaluate the upstream propagation of gravity weaves in our simulations using the Froude number, as done by Wu and Porté-Agel (2017). The Froude number characterizes the balance between flow acceleration or deceleration and the pressure gradient imposed by the displacement of the stably stratified flow $Fr = \overline{U}/\sqrt{g'H}$, where $\overline{U}$ is the boundary-layer bulk velocity, $g' = g\Delta\theta/\theta_0$ is the reduced gravity accounting for the inversion strength, and $H$ is the boundary layer height. Wu and Porté-Agel (2017) suggest gravity waves amplify the blockage effect in subcritical flow ($Fr < 1$), where pressure distur-

bances propagate upstream. The Froude number in our weak and moderate stability simulations is 1.2 and 1.35, respectively, characteristic of supercritical flow ($Fr > 1$), thus pressure disturbances do not propagate upstream.

**Appendix D: Grid resolution**

Grid resolution in our simulations is sufficient to resolve most turbulence kinetic energy across the turbine rotor layer (Figure D1). For the non-linear backscatter and anisotropy subgrid-scale turbulence model (Kosović, 1997), the total turbulence kinetic

energy $\overline{k}_{tot}$ is given as $\overline{k}_{tot} = \frac{1}{2}(\overline{u_i'u_i'} + m_{ii}) + \overline{k}_{SGS}$, where $\overline{u_i'u_i'}$ represents the resolved TKE, $m_{ii}$ are the normal subgrid-scale stress components, and $\overline{k}_{SGS}$ is the subgrid-scale TKE. Nearly 80% of TKE in the turbine rotor layer is resolved by the numerical grid for both simulations (Figure D1). Because less than 80% of TKE in the lower rotor layer is resolved in the weak stability case ($\overline{k}_{res}/\overline{k}_{tot} = 0.78$ at $z = 30\,\mathrm{m}$), a finer grid is used for the simulation of moderately stably stratified flow.

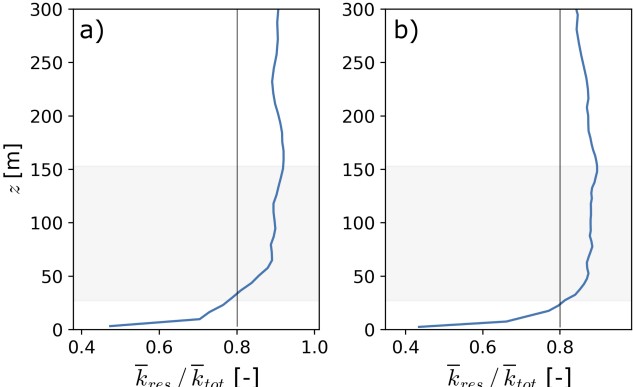

**Figure D1.** Fraction of resolved TKE in the surface layer for the weak (a) and moderate (b) stability cases. The solid, black line corresponds to 80% of resolved TKE. The grey shaded area corresponds to the turbine rotor layer.

*Author contributions.*  MSG had the lead on modeling, data analysis, and paper writing. JKL provided general guidance, and contributed to

the data analysis and in reviewing the paper. JDM provided suggestions with the modeling framework and contributed to reviewing the paper. RSA provided guidance on the data analysis and contributed to reviewing the paper.

*Competing interests.* One of the co-authors is an associate editor for Wind Energy Science.

*Acknowledgements.* This work was supported by an agreement with NREL under APUP UGA-0-41026-125. This work was authored in part by the National Renewable Energy Laboratory, operated by Alliance for Sustainable Energy, LLC, for the U.S. Department of Energy (DOE) under Contract No. DE-AC36-08GO28308. Funding provided by the U.S. Department of Energy Office of Energy Efficiency and Renewable Energy Wind Energy Technologies Office. The views expressed in the article do not necessarily represent the views of the DOE or the U.S. Government. The U.S. Government retains and the publisher, by accepting the article for publication, acknowledges that the U.S. Government retains a nonexclusive, paid-up, irrevocable, worldwide license to publish or reproduce the published form of this work, or allow others to do so, for U.S. Government purposes. JDM and RSA contributed under the auspices of the U.S. Department of Energy (DOE) by Lawrence Livermore National Laboratory, under contract DE-AC52-07NA27344.

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
