# Peer review of "Investigating the physical mechanisms that modify wind plant blockage in stable boundary layers"

_Wind Energy Science, 2023_

## Referee Comment (RC1)

The paper uses two WRF-LES simulations of a large, generic wind farm to investigate how and why wind-farm blockage varies with surface layer stability. The underlying physical mechanisms are explored based on a detailed analysis of the streamwise momentum budget components. Interestingly, the paper shows that the adverse pressure gradient upstream of a front-row turbine is nearly identical to the pressure gradient upstream of a standalone turbine, and the difference between single-turbine induction and wind-farm blockage stems from the vertical momentum advection. The paper is well-written and has a clear structure. I appreciate that the paper has one clear research objective, and accordingly the analysis of the wind farm simulations focuses solely on the upstream flow behaviour in order to address the research question. I am a bit puzzled by the claim that this paper investigates blockage in the absence of gravity waves, and I do have some related questions about the numerical setup. Please find below a list of comments and suggestions.

**Main comments**

1. One line 66, the authors claim that they investigate wind-farm blockage in the absence of gravity waves. How exactly do you ensure that there are no gravity waves in your simulation? I think this is quite a significant assumption and should therefore be discussed in more detail.

2. The LES uses a two-domain configuration with one-way nesting. Can you give more details about the domain nesting? How large is the outer domain compared to the inner domain? What boundary conditions are imposed on the inner domain? If this is an inflow-outflow type domain, is there a transitional period to impose (blend) the inflow wind speed? What is the outlet boundary condition (simple outlet condition or again blending towards the parent solution)?

3. How is the grid resolution chosen? The authors mention that a finer grid is used in the more stable case, but how did you determine that the employed grid resolution is sufficient?

4. An implicit Rayleigh damping layer of 1000 m is used to avoid wave reflection. How do you know this leads to sufficient damping? Did you check for wave reflections? Other LES studies typically use Rayleigh damping layers of 10 km or more (see, e.g., work by Allaerts and Meyers, or Lanzilao and Meyers), so 1000 m seems quite small to me.

5. How does the power of the entire wind farm vary wind stability? I imagine this will also affect the amount of blockage.

6. Why do you call the net-upward vertical motion in the region upstream of the wind farm a secondary circulation? I don't fully understand why you see this as a circulation (any similarities with other flow scenarios?). Note that this upward flow displacement has been noted by others in the past (see, e.g., Allaerts and Meyers 2017).

7. Line 265 "Larger vertical shear of the horizontal velocity in the moderate stability case contributes to the increased vertical momentum advection compared to the weak stability case." I think this is the most important finding of the paper, but it is not entirely clear to me how vertical shear affects vertical momentum advection. Can you please elaborate?

8. Appendix C: It is an interesting approach to assess the presence of gravity waves by means of the phase shift between pressure and vertical velocity signals (note that it is not clear at which height the signals are obtained, or are they averaged over heights?). However, I'm not entirely sure whether these phase relations still hold when you have wave reflections. When there are wave reflections, these lead to standing wave patterns and I can imagine that for those cases the phase relationships change. Did you look at vertical cross-sections of pressure and vertical velocity throughout the entire numerical domain?

**Other scientific comments**

1. Line 139-140 "The balance between turbulence production via shear below the LLJ's nose and temperature stratification result in a shallow stable boundary layer" and line 145-146 "The balance between turbulence production via shear below the capping inversion and temperature stratification result in a deep stable boundary layer." You seem to suggest two different physical mechanisms. Are you saying the stable boundary layer is shallow or deep depending on whether buoyant destruction is balanced by shear production below the LLJ or below the capping inversion? I don't think this is a proper description of what happens physically.

2. Line 143-144 "After 3 hr, temperature stratification close to the surface reduces the vertical transport of momentum above 400 m, and a LLJ starts forming (Figure 5f,g)." This goes a bit fast, please explain. How does the stratification close to the surface affect transport above 400m and lead to a LLJ?

3. Line 220-221 "Whereas the v-velocity transports momentum to both sides of the turbine, the w-velocity primarily transports momentum upwards." How do you know that w-velocity primarily transports momentum upwards? Did you check this? You can't make this conclusion only based on Figure 11.

4. Line 237 "Vertical momentum advection is the primary forcing mechanism ... ." Figure 13 shows that the pressure contribution is also significant. I agree that only the vertical momentum advection is affected by stability, but you make it sound like the pressure gradient force is insignificant. See also lines 297-298 "The pressure gradient force is also present upstream of the wind plant; however, the primary mechanism decelerating the flow is the vertical advection of horizontal momentum." I don't think this is true. I'd say the flow deceleration is due to the combined effect of vertical advection and pressure gradient force.

**Minor/technical comments**

1. Figure 3: Please mention how the effective grid resolution is determined. Is this from visual observation, or is there a certain method to calculate the effective grid resolution?

2. Line 122-126: "Large, localized gradients of the horizontal velocity instigate large-scale turbulence early in the simulation, which then cascades into small-scale turbulence (Figure 3)" and "Localized shear instabilities instigate turbulence throughout the boundary layer within the first hour of the simulation. These structures break up rapidly into smaller eddies, reducing shear until a quasi-steady state is reached. Turbulence structures form rapidly close to the surface and propagate upwards (Figure 3)." It sounds to me as if you are saying the same twice. Is this intentional or should one of the two phrases be removed? Furthermore, it is not clear how turbulence is initialized. Do you apply random perturbations to the velocity field?

3. Line 144 "Vertical turbulence redistributes ... ." Should this be vertical turbulent mixing? Not sure what you mean by "vertical turbulence".

4. Figure 5: It is hard to appreciate the formation of a LLJ from the evolution of the wind speed components. Wouldn't it be more informative to show the wind speed magnitude and the wind direction with height?

5. Line 179-180 "Horizontal wind speed is on average 31 % slower ... ." Do you see a 31% difference in the wind speed or in the wind speed deficit? This is a big difference, so please clarify.

6. Line 196-198: Where are the order of magnitudes of the various forces coming from? Did you get these values from the results or are they a simple order of magnitude estimate? Please specify.

7. Line 221-222 "The streamwise velocity advects momentum back into the induction region of the turbine" and line 235-236 "The streamwise momentum advection replenishes momentum upstream of the first turbine row." I find the formulation of these sentences a bit weird and therefore confusing. Streamwise momentum advection simply acts as a source of energy because the flow is decelerating. Please reformulate to make it more clear what you mean.

8. Line 240 "Momentum advection by the v-velocity is 10.1 % as large as the ... ." This confusing, I guess you are saying that the momentum advection by the v-velocity is only 10.1 % of the vertical momentum advection?

9. Discussion of Figure B1 is a bit hard to follow. What does a negative value for the pressure gradient force mean? Does a negative value correspond to an upward or downward pressure force?

---

## Referee Comment (RC3)

**Paper title:** "Investigating the physical mechanism that modify wind plant blockage in stable boundary layers"

The authors simulated an idealized wind farm on flat terrain operating in two different stable boundary layers. The simulations were run back-to-back with a turbine operating in isolation. The results of these simulations were then analyzed to quantify the impact of blockage on the flow upstream of the wind farm and also on the upstream row of turbines. Further, the authors interrogated the solutions to understand the physical mechanisms behind these blockage effects.

If accepted for publication, I think this paper would be a strong addition to the growing literature on wind farm blockage effects. The analysis of the simulated flow is lucid and compelling. The main findings, in my view, are important and mostly new.

The main area where I feel the paper could be improved is putting the research into context. The next section of this review focuses on this aspect. I include a number of miscellaneous comments after the context section.

**Putting the findings into context:**

**Gravity waves** – Gravity waves are referred to a number of times in the manuscript, mostly in the vein of we-are-not-dealing-with-gravity-waves-in-this-study. The introduction puts this explicitly: "Here, we investigate how atmospheric stability modifies upstream blockage in the absence of gravity waves." There is even a section in the appendix substantiating the claim that there are no gravity waves in the flow solutions. I do not think this approach to setting the no-gravity-wave stage is necessary and may even be counterproductive, potentially leading readers to believe that the findings are limited to gravity-wave free conditions.

There are good reasons to believe wind-farm-induced gravity waves are very common. The manuscript implies the findings in the two Bleeg papers are not affected by gravity waves (see line 50), but this is not the case. All our wind farm simulations have gravity waves, at least to some degree. Gravity waves, as you know, are the primary means by which disturbances are transmitted in density-stratified flow. Such a disturbance might be an obstacle like a wind farm causing flow to rise. The resulting gravity waves—or speaking more generally, inviscid effects related to disturbed stratified flow—and their effect on the wind farm will depend upon the characteristics of the stratification and the wind farm itself. The authors appear to be simulating a set of conditions where the effects of gravity waves on flow upstream of the wind farm is not as pronounced as in the first Allaerts and Meyers JFM paper. I think it is sufficient to say just that.

There is strong numerical evidence at this point that inviscid effects related to stratification above the boundary layer—and the horizontal pressure gradients associated with them—can have a significant impact on blockage effects. This paper helps bring to light another important influence on the production of leading row turbines relative to what they would produce in isolation: the vertical advection of u-momentum. The finding not only improves our physical understanding of blockage effects, but also has significant implications for the modelling of these effects.

**Engineering models for blockage** – The engineering models designed to predict blockage effects are inviscid, potential flow-type models, which do not account for shear and therefore are not able to reliably account for what this paper suggests is one of the most important contributors to front row blockage loss: the vertical advection of u-momentum. It is up to the authors, but I think it might be worth emphasizing this in the paper.

The manuscript states that blockage is not currently accounted for in EYA's. This was true in 2018, but it is not true anymore. Almost all EYA's account for blockage in one way or another. And in many cases, a potential flow model is used to account for the blockage effect. Potential flow models, as discussed, will miss a significant contributor to the impact of blockage on turbine production. I feel that the wind industry community would benefit from having this point highlighted.

**Has anyone looked into this before?** According to the manuscript, the dominant mechanism causing front row wind turbines to produce less than they would in isolation is the vertical advection of u-momentum (at least in these simulations). To me, this is the most important finding in the paper. The Discussion and conclusion sections highlights others who have focused on adverse pressure gradients as a key driver behind blockage, and then explains that vertical advection of u-momentum amplifies the impact on the front row turbines. I'm a bit biased here, but it is worth mentioning that Bleeg and Montavon included a full section on this subject? The section makes the point that the combination of shear and flow rising as it approaches the wind farm, due to the presence of the wind farm and the ground, "appears to be an important factor in determine the magnitude of the blockage loss." Immediately following, the paper reads, "in case 9, for example, the streamline passing through the hub in the wind farm configuration originates approximately 4.5 m below the hub-intersecting streamline in the isolated case. In turn, the wind speed on the streamline far upstream is approximately 2% lower in the wind farm configuration compared to isolated operation. This significant wind speed difference is the result of the vertical flow deflection combining with the increased shear that prevails in stable conditions."

In my opinion, the physical explanation provided in the manuscript under review is more clear, complete, and convincing that what we provide in Bleeg and Montavon. Much credit is due to the authors for this important finding. That said, I think it is fair to say that the Bleeg and Montavon paper did highlight the important influence of shear in combination with the upstream vertical deflection of flow as it relates to the impact of blockage on the production of leading row turbines relative to a turbine in isolation.

I am not an academic and am not familiar with what is required in a situation like this (also, I suppose I am not without bias in this regard), so I leave it to the authors and the editor to decide whether this should be acknowledged in the paper. For what it is worth, I think referencing the earlier result could further strengthen the credibility of the current finding. A sentence or two would be sufficient. And if it is followed by something indicating that your analysis more complete, I would not object, because it is.

**Miscellaneous:**

The following questions, comments, and suggestions are in roughly order of priority

   A. I can't quite tell where the overall control volume in Figure 13 ends in the x-direction. The caption says it is bounded at the first turbine row (x = 5670 m). Clear enough, but where is that location in Figure 12? Is it at 0 D or just upstream? The reason why I ask is that that thrust body forces from the GAD are being applied to cells that include locations just upstream of 0 D, resulting in a rapid drop in pressure. If the end of the control volume in Figure 13 does correspond to 0 D, how would the results change if the end of the control volume were moved to just in front of the GAD?
   B. Again, I found the analysis of the results in sections 3 and 4 to be clear and convincing. That said I wonder if you could go just a little further to provide more physical insight and help the reader connect the dots towards your key finding. I refer specifically to the significant influence of the vertical advection of u-momentum. Perhaps you could break this down a bit.

It could help drive home the point of the importance of shear. The x-component of velocity is clearly higher at the top of the control volume than the bottom. I'm not sure how the vertical component of velocity varies streamwise upstream of the wind farm, but I suspect it is positive and generally increases as flow approaches the wind farm. Of course, the trend may differ between the top and bottom of the control volume (in fact, the vertical component of velocity may be negative close to the rotors). I think it would be nice to have these things related to the vertical advection of u-momentum broken down, though I concede that what you already have in the report is sufficient to make your point.

C.  This is a big statement in the paper: "Given that the normalized pressure gradient force remains unchanged with atmoshperic stability and turbine array size, differences in blockage are caused by momentum redistribution in the induction region." Firstly, you'll want to correct the misspelling of atmospheric. Secondly, I just want to put this into context. In practice, when evaluating the impact of blockage or wakes on a wind farm, what we care about is the power production of the wind farm turbines relative to what they each would produce in isolation. In other words, we care about the wind conditions that each turbine experiences relative to the conditions it would experience in isolation. My interpretation of your analysis is that, at least for the simulated conditions, the vertical advection of u-momentum is *the dominant factor* affecting the production of leading row turbines relative to what they would produce in isolation. It is by far the main physical mechanism behind blockage loss for these turbines—again, for the simulated conditions. Am I interpreting your work correctly?

D.  I'm not sure your simulation setup can reliably capture gravity waves. With our own steady-state RANS model, a domain height of 3500 m (too low) and a damping layer thickness of 1000 m (too thin) would yield significantly different results with respect to gravity waves than our standard domain (top boundary at 17,000 m and much thick damping layer). That said, if you were to re-run your analysis with a much larger domain, I doubt your main findings would be much different. Thus, in my view, it is not required to run this sensitivity check, though it would be a nice-to-have. If future studies focus on gravity waves—and more broadly the influence of the stably stratified atmosphere above the boundary layer—such a sensitivity study would be needed.

E.  As suggested above, I would consider just dropping Appendix C. However, if you keep it, could you please clarify the height at which the values in Figure C1 are being plotted?

F.  If you pursue this research further, it would be interesting to know what you find when simulating a neutral boundary layer and/or an unstable boundary layer

---

## Author Response (AR1)

**Summary of changes:**

Dear editor,

Changes to the manuscript include:

1. Clarification of the physical mechanisms that modify wind plant blockage in our simulations.
2. More detailed discussion on gravity waves in our domain.
3. Clarification on the control volumes used for each analysis.
4. Contextualization of our results within the scientific literature.
5. More detailed literature review.
6. Power production of front-row turbines.
7. Additional analysis on the induction region of a stand-alone and single front-row turbine in the wind plant.
8. Clarification of the numerical setup (grid resolution, parent-nest domains, boundary conditions).
9. Clarification in captions of figures.

Some responses below are very similar because they address the same issues. We include the response for each reviewer.

**Response to Referee #1: Dries Allaerts**

Authors' responses to reviewer comments appear in blue text. Line numbers referenced in the authors' responses refer to the revised document. Figures with Arabic numerals (e.g., Figure 10) correspond to the revised manuscript, Figures with roman numerals (e.g., Figure iv) only appear on the response to reviewer's comments.

The paper uses two WRF-LES simulations of a large, generic wind farm to investigate how and why wind-farm blockage varies with surface layer stability. The underlying physical mechanisms are explored based on a detailed analysis of the streamwise momentum budget components. Interestingly, the paper shows that the adverse pressure gradient upstream of a front-row turbine is nearly identical to the pressure gradient upstream of a standalone turbine, and the difference between single-turbine induction and wind-farm blockage stems from the vertical momentum advection. The paper is well-written and has a clear structure. I appreciate that the paper has one clear research objective, and accordingly the analysis of the wind farm simulations focuses solely on the upstream flow behaviour in order to address the research question. I am a bit puzzled by the claim that this paper investigates blockage in the absence of gravity waves, and I do have some related questions about the numerical setup. Please find below a list of comments and suggestions.

We thank the reviewer for providing thoughtful comments that helped improve our manuscript.

Main comments

1. One line 66, the authors claim that they investigate wind-farm blockage in the absence of gravity waves. How exactly do you ensure that there are no gravity waves in your simulation? I think this is quite a significant assumption and should therefore be discussed in more detail.

Thank you for highlighting this because it is an area of active research. We clarify throughout the entire manuscript that we investigate blockage with minimal upstream propagation of gravity waves (this was also requested by another reviewer) and direct the reader to the appendix as follows:

Line 68: "Here, we investigate how atmospheric stability modifies upstream blockage with minimal upstream propagation of gravity waves (see Appendix C for a discussion on gravity waves in our domain)"

We also clarify the discussion on gravity waves (see Main Comment #8 below) and include the energy associated with wave reflection from the model top (see Main Comment #4 below) as follows:

Line 483: "Spurious waves can sometimes modify the correlation between atmospheric variables in upstream-propagating gravity waves (Lanzilao and Meyers, 2022). Because the only potential source of gravity waves in our simulations is in the boundary layer (i.e., the wind plant), then waves with a downward group velocity (positive phase speed) and outside the boundary layer must be due to spurious reflections (Taylor and Sarkar, 2007). We quantify wave reflection following the methodology outlined in Taylor and Sarkar (2007). We find that 7.1% and

5.8% of the total vertical kinetic energy $0.5w'^2$ is associated with downward energy propagation for the weak and moderate stability cases, respectively, which is comparable to the wave reflection reported in other studies (Taylor and Sarkar, 2007; Allaerts and Meyers, 2018, 2017)."

2.  The LES uses a two-domain configuration with one-way nesting. Can you give more details about the domain nesting? How large is the outer domain compared to the inner domain? What boundary conditions are imposed on the inner domain? If this is an inflow-outflow type domain, is there a transitional period to impose (blend) the inflow wind speed? What is the outlet boundary condition (simple outlet condition or again blending towards the parent solution)?

We add clarification about the size of the parent domain and one-way nesting in the manuscript as follows:

Line 88: "We use a two-domain configuration with flat terrain to evaluate the blockage effect from wind plants. A periodic LES domain provides the boundary conditions for a nested LES domain via one-way nesting (i.e., atmospheric conditions for the outermost grid cells in the nested domain are specified from the parent domain)."

Line 96: "The parent domain is 10 grid points larger than the nest in the horizontal directions."

We investigate the effect from prescribed boundary conditions by considering the velocity field close to the domain boundaries. We find that boundary conditions have minimal effects on the flow close to the wind plant (Figure i). The velocity field in the induction region asymptotes to the velocity at the upstream end of the domain. The velocity at x = -30D in the nested domain is virtually the same to the velocity of the parent domain (vertical black line in Figure i). At the downstream end of the domain, the wake recovery only appears to be influenced by the boundary conditions very close to the domain boundary.

[Figure]

*Figure i: Time-averaged velocity field for the parent and nested domain. The x-axis is re-scaled to locate 0 at the first turbine row.*

Wake recovery varies for each turbine row (Figure ii). Wake recovery is faster for the rows in the trailing edge of the wind plant. The change in wake recovery between third and fourth rows is comparable to the change in wake recovery between the second and third rows. Furthermore, the velocity field only displays a sudden change very close to the domain outflow (Row 4). The horizontal velocity rapidly increases 15D downstream of the last row of the wind farm and the domain boundary is 15.7D downstream of the last turbine row.

[Figure]

*Figure ii: Wake recovery downstream of each turbine row of the wind plant. The dashed red line illustrates the distance downstream of the last turbine row where the velocity is likely influenced by the boundary conditions.*

Because one-way nesting is regularly used for boundary-layer simulations of wind turbines (Mirocha et al., 2014; Aitken et al., 2014; Arthur et al., 2020; Sanchez Gomez et al., 2022; Wise et al., 2022) and because the velocity field is only minimally affected by boundary conditions close to the domain boundaries, we decide not to include this detailed information in the manuscript. We rather add clarification to the text as follows:

Line 107: "As will be shown later in the manuscript, the velocity deceleration in the induction region is virtually zero 30D upstream of the wind plant. Therefore, 45D of fetch upstream of the wind plant is deemed sufficient to investigate the induction region of the turbine array."

Furthermore, we adjust the x-axis in Figure 8 to show the velocity deceleration far upstream as follows:

Line 183: "The velocity deceleration asymptotes to zero far upstream (x<-30D) for both atmospheric conditions (Figure 8)."

[Figure]

Figure 8: Normalized velocity deficit $\left(\Delta U = \frac{\overline{U}-\overline{U}_\infty}{\overline{U}_\infty}\right)$ for the inter- and intra-turbine regions upstream of the wind plant for each atmospheric condition…

3. How is the grid resolution chosen? The authors mention that a finer grid is used in the more stable case, but how did you determine that the employed grid resolution is sufficient?

Because another reviewer also asked about grid resolution, we now include an additional appendix in our manuscript:

Line 493: Appendix D

"Grid resolution in our simulations is sufficient to resolve most turbulence kinetic energy across the turbine rotor layer (Figure D1). For the non-linear backscatter and anisotropy subgrid-scale turbulence model (Kosović, 1997), the total turbulence kinetic energy $\overline{k}_{tot}$ is given as $\overline{k}_{tot} = \frac{1}{2}\left(\overline{u_i'u_i'} + m_{ii}\right) + \overline{k}_{SGS}$, where $\overline{u_i'u_i'}$ represents the resolved TKE, $m_{ii}$ are the normal subgrid-scale stress components, and $\overline{k}_{SGS}$, is the subgrid-scale TKE. Nearly 80% of TKE in the turbine rotor layer is resolved by the numerical grid for both simulations (Figure D1). Because less than 80% of TKE in the lower rotor layer is resolved in the weak stability case ($\overline{k}_{res}/\overline{k}_{tot}$= 0.78 at z = 30 m), a finer grid is used for the simulation of moderately stably stratified flow."

[Figure]

Figure D1: Fraction of resolved TKE in the surface layer for the weak (a) and moderate (b) stability cases. The solid, black line corresponds to 80% of resolved TKE. The grey shaded area corresponds to the turbine rotor layer.

4. An implicit Rayleigh damping layer of 1000 m is used to avoid wave reflection. How do you know this leads to sufficient damping? Did you check for wave reflections? Other LES studies typically use Rayleigh damping layers of 10 km or more (see, e.g., work by Allaerts and Meyers, or Lanzilao and Meyers), so 1000 m seems quite small to me.

We quantify the energy reflected from the domain top following the methodology from Taylor and Sarkar (2007). Because the only potential source of gravity waves is in the boundary layer, then waves with a downward group velocity (positive phase speed) and outside the boundary layer must be due to spurious reflections (Taylor and Sarkar, 2007).

We quantify wave reflection by observing the $w'$ field in a frame moving with the geostrophic wind, as in Taylor and Sarkar (2007). Then, we transform $w'$ into the frequency and wavenumber domain. We decompose the spectrum into upward and downward propagating waves for vertical levels in between the top of the inversion layer and the bottom of the damping layer. Finally, an inverse Fourier transform yields the variable $w'$ in physical space for internal waves with downward energy propagation. We find that 7.1% and 5.8% of the total vertical

kinetic energy $\left(\frac{1}{2}\overline{w'^2}\right)$ is associated with downward energy propagation for the -0.2 K/h and -0.5 K/h simulations, which is comparable to the wave reflection seen in other studies (Taylor and Sarkar, 2007; Allaerts and Meyers, 2017, 2018).

We include the relevant information in Appendix C, as reported in Main Comment #1.

5. How does the power of the entire wind farm vary with stability? I imagine this will also affect the amount of blockage.

Thank you for your suggestion, we expand our analysis on turbine power as follows:

Line 195: "Even though the wind speed slowdown from blockage is small, front-row turbines in the wind plant produce on average 5.2% less power than a stand-alone turbine (Figure 10). Because winds are slightly faster in the moderate stability case compared to the weak stability case, turbine power is also expected to differ. As a result, we evaluate the difference in power production between the turbines in the wind plant and a stand-alone turbine for the same atmospheric conditions. Just as the velocity deceleration is modified with atmospheric stability, turbine underperformance is more severe in the moderate stability case compared to the weak stability case. Whereas turbines in the first row produce on average 4% less power than a stand-alone turbine for the weak stability condition, front-row turbines produce on average 6.5% less power than a stand-alone in the moderate stability case. Downstream of the first row of the wind plant, turbine power is primarily dominated by the evolution of the wake. Turbine wakes persist longer in stable boundary layers because of reduced turbulence mixing (Dörenkämper et al., 2015; Lee and Lundquist, 2017), so we expect downstream turbines to produce less power in the moderate stability case compared to the weak stability case."

[Figure]

Figure 10: Normalized turbine power for each row of the wind plant and each atmospheric condition. The mean turbine power for the i-th row of the wind plant $\overline{P}_i$ is normalized over the mean turbine power of a stand-alone turbine $\overline{P}_{st}$.

We also want to emphasize that the momentum fluxes presented in our manuscript are normalized to account for discrepancies in inflow conditions. We update the manuscript as follows:

Line 230: "The u-momentum flux at the inflow of the control volume $\mathcal{V}$ in Figure 11 is larger in the moderate stability case compared to the weak stability case due to slightly faster hub-height winds. Consequently, the magnitude of the momentum fluxes and turbine power is expected to be larger in the moderate stability case as well. To contrast the momentum balance between different atmospheric stability conditions and turbine array sizes, we normalize the forcing terms in Eq. 2 using the momentum flux at the inflow of the control volume far upstream $(\rho \bar{u}_\infty \bar{u}_\infty S_x)$ for each stability case."

6.  Why do you call the net-upward vertical motion in the region upstream of the wind farm a secondary circulation? I don't fully understand why you see this as a circulation (any similarities with other flow scenarios?). Note that this upward flow displacement has been noted by others in the past (see, e.g., Allaerts and Meyers 2017).

We replace "secondary circulation" with "secondary flow feature" over the entire manuscript, as suggested by the reviewer. As an example, line 293 now reads as: "Mass balance indicates the slowdown of the u-velocity in the turbine rotor layer $(\Delta(\rho \bar{u} S_x) < 0)$ is balanced by the development of a secondary flow feature in the form of net-upwards vertical motion $(\Delta(\rho \bar{w} S_z) > 0)$ for both stability conditions (i.e., $\Delta(\rho \bar{u} S_x) + \Delta(\rho \bar{w} S_z) \approx 0)$."

We also comment on findings from other studies than mention the vertical deflection of the flow:

Line 345: "Other simulation studies have also noted this vertical deflection of the flow (e.g., Wu and Porté-Agel, 2017; Allaerts and Meyers, 2017)."

7.  Line 265 "Larger vertical shear of the horizontal velocity in the moderate stability case contributes to the increased vertical momentum advection compared to the weak stability case." I think this is the most important finding of the paper, but it is not entirely clear to me how vertical shear affects vertical momentum advection. Can you please elaborate?

Thank you for this comment. Indeed, this is one of the most important findings. We added clarification in the manuscript as follows:

Line 300: "The vertical velocity advects horizontal momentum out of the turbine rotor layer (Figure 19). Vertical advection of horizontal momentum is 20% larger in the moderate stability case compared to the weak stability upstream of the first turbine row (Figure 19b). Larger vertical shear of the horizontal velocity in the moderate stability case compared to the weak stability case is the primary cause for the increased vertical advection of horizontal momentum. Shear $\left(\frac{\Delta \bar{u}}{\Delta z} = \frac{\bar{u}_t - \bar{u}_b}{D}\right)$ between the bottom $\bar{u}_b$ and top $\bar{u}_t$ of the turbine rotor layer is 43.6% larger in the -0.5 K/h simulation compared to the -0.2 K/h simulation. Similarly, the vertical velocity in the turbine rotor layer is 20% larger in the moderate stability case than in the weak stability case between x = -6D and x = 0D. The vertical velocity is expected to be larger in the moderate stability case because, as shown in Figure 18, the streamwise slowdown of the flow is transformed almost entirely into vertical motions. As a result, advection of horizontal momentum by the vertical velocity $\Delta(\rho \bar{u} \, \bar{w} \, S_z) = \rho S_z(\bar{u}_t \bar{w}_t - \bar{u}_b \bar{w}_b)$ is amplified."

8.  Appendix C: It is an interesting approach to assess the presence of gravity waves by means of the phase shift between pressure and vertical velocity signals (note that it is not clear at which height the signals are obtained, or are they averaged over heights?).

However, I'm not entirely sure whether these phase relations still hold when you have wave reflections. When there are wave reflections, these lead to standing wave patterns and I can imagine that for those cases the phase relationships change. Did you look at vertical cross-sections of pressure and vertical velocity throughout the entire numerical domain?

You bring up an interesting point about spurious gravity waves, which have been shown to distort the pressure and vertical velocity fields in (Lanzilao and Meyers, 2022). However, as we point out in Main Comment #4, we have marginal energy being reflected from the model top and therefore we do not expect the phase shift to be driven by spurious waves. Furthermore, as shown by Lanzilao and Meyers (2022), spurious gravity waves also cause artificial oscillations to the pressure and vertical velocity, which are not present in our simulations.

We examined the vertical cross-sections of the pressure and vertical velocity over the entire domain (Figure iii), and we observe the same trend: the pressure and vertical velocity are out of phase. For example, the maximum in vertical velocity occurs at the location of the first turbine row (x = 0D), whereas the perturbation pressure at x=0D is zero. However, we decided to include the line plot because it distinctively shows the maxima/minima for each variable and where each variable intersects $y = 0$. By clearly showing this behavior, the line plots visibly depict the phase shift between the atmospheric variables. The line plot also illustrates that we do not have spurious waves, which can cause artificial oscillations in the pressure field at the height of the inversion layer (Lanzilao and Meyers, 2022).

[Figure]

*Figure iii. Vertical cross section of the time-averaged vertical velocity (left), pressure perturbation (middle) and potential temperature (right) fields for the -0.5 K/h simulation. Each variable is normalized and therefore non-dimensional. The dashed horizontal line in the left panel shows the height of the damping layer.*

Thank you for pointing out that we do not include the height for the data shown in the line plots. We added clarification in the caption of Figure C1 in the manuscript and updated the Appendix C as follows:

Lines 467: "Figure C1 shows the streamwise evolution of the deviation in vertical velocity, pressure and potential temperature from the inflow of the domain at the capping inversion."

Figure C1: "Streamwise evolution of the vertical velocity, pressure and potential temperature deviation from inflow conditions for the weak (a) and moderate (b) stability cases at z = 1200 m. Each variable $a_i$ is normalized as $\hat{\bar{a}}_i = \dfrac{a_i}{\max(a_i) - \min(a_i)}$ and averaged along the y-direction (from y = 1953 m to y = 5922 m). The gray shaded area in each panel represents the region covered by the wind plant."

Line 481: "Note that we show the phase shift between the pressure, vertical velocity and potential temperature above the capping inversion; however, this phase shift is also observed in the boundary layer."

Other scientific comments

1. Line 139-140 "The balance between turbulence production via shear below the LLJ's nose and temperature stratification result in a shallow stable boundary layer" and line 145-146 "The balance between turbulence production via shear below the capping inversion and temperature stratification result in a deep stable boundary layer." You seem to suggest two different physical mechanisms. Are you saying the stable boundary layer is shallow or deep depending on whether buoyant destruction is balanced by shear production below the LLJ or below the capping inversion? I don't think this is a proper description of what happens physically.

We restructure and shorten this paragraph because it was confusing. This paragraph now reads as:

Line 146: "Boundary layer evolution varies with surface forcing (Figure 4). A fast cooling rate (i.e, -0.5 K/h) produces increasing temperature stratification below 400 m and quasi-neutral stratification up to the capping inversion (Figure 4c). The rapid development of a stable layer close to the surface reduces the vertical transport of momentum, suppressing turbulence aloft. A broad low-level jet (LLJ) develops after 4 hr as turbulence aloft decreases (Figure 4a,b). Boundary-layer evolution for the slower cooling rate is slightly different. A -0.2 K/h produces increasing temperature stratification up to the capping inversion (Figure 4h). The slow cooling rate initially produces nearly uniform cooling of the entire turbulent layer. After 3 hr, temperature stratification close to the surface is large enough to reduce the vertical transport of momentum and a LLJ starts forming close to the capping inversion (Figure 4f,g). Because of a slower cooling rate, the gradual reduction in vertical turbulent mixing results in a deeper boundary layer in the weak stability case compared to the moderate stability case."

2. Line 143-144 "After 3 hr, temperature stratification close to the surface reduces the vertical transport of momentum above 400 m, and a LLJ starts forming (Figure 5f,g)." This goes a bit fast, please explain. How does the stratification close to the surface affect transport above 400m and lead to a LLJ?

Please see comment above.

3. Line 220-221 "Whereas the v-velocity transports momentum to both sides of the turbine, the w-velocity primarily transports momentum upwards." How do you know that w-velocity primarily transports momentum upwards? Did you check this? You can't make this conclusion only based on Figure 11.

We appreciate the reviewer's attention to detail. Figure iv shows that the vertical velocity immediately upstream of the turbine can be negative at the bottom of the turbine rotor layer. We

decide not to include this figure in the text because it is a minor detail that deviates from the main objective of the manuscript.

[Figure]

*Figure iv: Time-averaged vertical velocity at the bottom (z = 27 m) of the turbine rotor layer for each atmospheric condition. The vertical velocity is averaged in the y-direction over the span of the wind plant.*

We update the manuscript as follows:

Line 251: "Whereas the v-velocity transports momentum to both sides of the turbine, the w-velocity primarily transports momentum upwards. Immediately upstream of the turbines (-1D<x<0D), the vertical velocity is negative at the bottom of the turbine rotor layer (not shown), transporting momentum downwards. Nonetheless, the vertical velocity is positive over the rest of the induction region, transporting momentum upwards."

4. Line 237 "Vertical momentum advection is the primary forcing mechanism ... ." Figure 13 shows that the pressure contribution is also significant. I agree that only the vertical momentum advection is affected by stability, but you make it sound like the pressure gradient force is insignificant. See also lines 297-298 "The pressure gradient force is also present upstream of the wind plant; however, the primary mechanism decelerating the flow is the vertical advection of horizontal momentum." I don't think this is true. I'd say the flow deceleration is due to the combined effect of vertical advection and pressure gradient force.

We agree with the reviewer that the pressure gradient has a strong contribution to wind plant blockage. We soften the language in our manuscript as follows:

Line 272: "The vertical advection of streamwise momentum and adverse pressure gradient are the primary forcing mechanisms influencing wind plant blockage in our simulations."

Line 338: "The pressure gradient force is also present upstream of the wind plant; however, the vertical advection of horizontal momentum contributes more to the deceleration of the flow."

Minor/technical comments

1. Figure 3: Please mention how the effective grid resolution is determined. Is this from visual observation, or is there a certain method to calculate the effective grid resolution?

We did not calculate the effective grid resolution of our simulations explicitly. Rather, we cite the expected effective grid resolution based on the reduced advection scheme. Our main goal is to

illustrate the scales at which we expect to resolve turbulence in our simulations. We update the caption of Figure 2 as follows:

Caption Figure 2: "Compensated turbulence spectra of the w-velocity for the $\Delta x = 7$ m neutral LES at z=90 m (a), z=300 m (b), and z=800 m (c). Spectra are color coded in 10-minute time increments since initialization. The dotted, black vertical line in each plot represents the effective grid resolution (4-5$\Delta x$) expected from the reduced advection scheme in our simulations (Kosović et al., 2016). The theoretical −2/3 Kolmogorov slope for the inertial range is indicated by the solid black line in each plot."

2.  Line 122-126: "Large, localized gradients of the horizontal velocity instigate large-scale turbulence early in the simulation, which then cascades into small-scale turbulence (Figure 3)" and "Localized shear instabilities instigate turbulence throughout the boundary layer within the first hour of the simulation. These structures break up rapidly into smaller eddies, reducing shear until a quasi-steady state is reached. Turbulence structures form rapidly close to the surface and propagate upwards (Figure 3)." It sounds to me as if you are saying the same twice. Is this intentional or should one of the two phrases be removed? Furthermore, it is not clear how turbulence is initialized. Do you apply random perturbations to the velocity field?

Thank you for pointing out duplicate information. We remove the first sentence in our manuscript. In addition, we include the following information in the Methodology section:

Line 128: "Furthermore, we speed up turbulence development by adding $\pm 0.5$ K perturbations to the potential temperature field below the capping inversion at initialization."

3.  Line 144 "Vertical turbulence redistributes ... ." Should this be vertical turbulent mixing? Not sure what you mean by "vertical turbulence".

We replaced vertical turbulence with vertical turbulence mixing in the manuscript.

4.  Figure 5: It is hard to appreciate the formation of a LLJ from the evolution of the wind speed components. Wouldn't it be more informative to show the wind speed magnitude and the wind direction with height?

Thank you for your suggestion, we modify the figure to include the magnitude of the horizontal velocity and the wind direction rather than the u- and v-wind components. We also modify Figure 6 to include the horizontal wind speed and wind direction rather than the u- and v-wind components.

5.  Line 179-180 "Horizontal wind speed is on average 31 % slower ... ." Do you see a 31% difference in the wind speed or in the wind speed deficit? This is a big difference, so please clarify.

We added clarification to mention the differences in velocity deficit as follows:

Line 192: "The horizontal wind speed deficit is 31% slower …"

6. Line 196-198: Where are the order of magnitudes of the various forces coming from? Did you get these values from the results or are they a simple order of magnitude estimate? Please specify.

We estimate the order of magnitude of the forces from data in our simulations. We added clarification in the text to stress that the order of magnitudes for each term is for our simulations specifically as follows:

Line 212: "We evaluate the balance between momentum advection by the mean flow and a pressure gradient. Even though the Coriolis force in our simulation domain is not negligible, Coriolis forcing in the induction region is small. The Coriolis parameter scales as $f_c \sim 10^{-4}\ s^{-1}$ and the v-velocity in the turbine rotor layer for both stability cases is on the order of $\bar{v} \sim 0.1\ m/s$, thus Coriolis forcing is of the order $f_c\bar{v} \sim 10^{-5}\ m/s^2$. Turbulence momentum redistribution is also small in the induction region of the wind plant for our simulations $\nabla \cdot (\overline{u'u'}) \sim 10^{-4}\ m/s^{-2}$. In comparison, momentum advection by the mean flow in the induction region is of the order $10^{-1}\ m/s^{-2}$ in our simulations."

7. Line 221-222 "The streamwise velocity advects momentum back into the induction region of the turbine" and line 235-236 "The streamwise momentum advection replenishes momentum upstream of the first turbine row." I find the formulation of these sentences a bit weird and therefore confusing. Streamwise momentum advection simply acts as a source of energy because the flow is decelerating. Please reformulate to make it more clear what you mean.

We reformulate both sentences to make them clearer:

Line 255: "The streamwise velocity replenishes momentum in the induction region as the flow decelerates."

Line 271: "The streamwise velocity replenishes momentum in the region upstream of the first turbine row $\Delta(\rho\bar{u}\,\bar{u}\,S_x)$."

8. Line 240 "Momentum advection by the v-velocity is 10.1 % as large as the ... ." This confusing, I guess you are saying that the momentum advection by the v-velocity is only 10.1 % of the vertical momentum advection?

We reformulate this sentence as follows:

Line 275: "Momentum advection by the v-velocity is only 10.1% (12.8%) of the vertical advection of u-momentum for the moderate (weak) stability case."

9. Discussion of Figure B1 is a bit hard to follow. What does a negative value for the pressure gradient force mean? Does a negative value correspond to an upward or downward pressure force?

Thank you for pointing this out. We include additional information in this section as follows:

Line 451: "Figure B1 shows the streamwise balance of vertical momentum. Because of the convention adopted throughout the manuscript ($\Delta X = X_{out} - X_{in}$ for an arbitrary variable $X$ on

the control volume $\mathcal{V}$ shown in Figure 11), negative terms in Figure B1 correspond to upward forces. As such, $\Delta \overline{p} S_z < 0$ is forcing the flow upwards and $\rho g' V > 0$ is forcing the flow downwards."

We also add clarification as follows:

Line 454: "Momentum balance for the w-velocity indicates the secondary flow (i.e., w-velocity) in the induction region is driven by a pressure gradient far upstream, and horizontal transport of w-momentum close to the turbines (Figure B1). Immediately upstream of the first turbine row (-1D<x<0D), horizontal advection of w-momentum drives upward motions. The sharp deceleration of the u-velocity immediately upstream of each front-row turbine ($\Delta \overline{u} < 0$ is large as shown in Figure 9) results in momentum replenishment, which is balanced by a downward pressure gradient force ($\Delta \overline{p} \, S_z > 0$). Farther upstream (x<-1D), an upward pressure gradient force ($\Delta \overline{p} S_z < 0$) overcomes buoyancy and the streamwise advection of vertical momentum. Redistribution of vertical momentum by the v- and w-velocity components is marginal within the induction region of the wind plant.

**Response to Anonymous Referee #2**

Authors' responses to reviewer comments appear in blue text. Line numbers referenced in the authors' responses refer to the revised document. Figures with Arabic numerals (e.g., Figure 10) correspond to the revised manuscript, Figures with roman numerals (e.g., Figure iv) only appear on the response to reviewer's comments.

The objective of the article is to investigate the mechanisms driving the development of the wind speed deceleration in front of wind farms responsible for the global blockage effect. To achieve this, the authors perform Large Eddy Simulations using WRF-LES. They compare simulations of two different atmospheric stability regimes (moderately stable and weakly stable) each with the actuator disk representation of a single turbine or a 10 x 4 turbines wind farm (NREL 5 MW).

The assessment of the physical effects driving global blockage is performed analyzing the different contributions to the steady state integral momentum equation for the u − velocity where the Coriolis force and turbulence contributions are neglected. A vertical momentum advection is identified as the main cause of global blockage.

The paper contributes to the currently increasing number of numerical investigations of the global blockage effect. Even though the findings that the vertical advection of momentum out of the farm inflow is correlated to global blockage (Strickland and Stevens, 2022), as well as the dependency of global blockage on atmospheric stability (Schneemann et al. 2021) is not new itself, the approach to separate the different contributions causing the flow deficit upstream a wind farm is novel and interesting. However, in the current draft the manuscript lacks a clear description of the interesting findings, compromising the achievement of the paper's objective. The paper is generally well written but needs corrections and clarifications detailed below. Further, a revised manuscript should better follow a storyline. The Figures are mainly clear and support the results, some changes are suggested in the following. We recommend to publish the paper after our major concerns and questions are addressed.

**Scientific comments**

**On the presentation of the physical mechanisms of global blockage**

The paper's objective is to clarify the fundamental physics of Global Blockage. However, the main findings are not highlighted well enough, and the argumentation towards the main results is hard to follow.

The different strengths in blockage comparing single turbine and wind farm resulting from different amounts of the flow being advected upwards (i.e. different vertical momentum transport), is one of the main findings of the paper, and it should be made clearer. The authors could e.g. display the different vertical wind speeds in front of the single turbine and the wind farm in a single plot.

We appreciate your suggestion. We now include this figure in the manuscript. Note that we only show the vertical velocity for the upper half of the turbine rotor layer because the vertical velocity can be negative at the bottom of the turbine rotor layer (Figure iv).

[Figure]

*Figure v: Time-averaged vertical velocity at the bottom (z = 27 m) of the turbine rotor layer for each atmospheric condition. The vertical velocity is averaged in the y-direction over the span of the wind plant.*

We now include the vertical velocity upstream of a front-row turbine in the middle of the wind plant and of a stand-alone turbine as follows:

Line 309: "Therefore, differences in vertical transport of horizontal momentum between a stand-alone turbine and a turbine in the wind plant are entirely due to the vertical velocity that forms upstream of the turbine array (Figure 21). For the wind plant, the secondary flow (i.e., net upwards w-velocity) extends farther upstream than for a stand-alone turbine (Figure 21)."

[Figure]

Figure 21: Streamwise evolution of the vertical velocity upstream of a stand-alone and front-row turbine in middle of the wind plant. The vertical velocity is averaged in the y-direction over the rotor diameter and in the z-direction over the top half of the rotor layer.

Another important finding is that the horizontal pressure gradient upstream of a turbine in isolation and a turbine in the first row of the farm is substantially equal. However, these results are very counterintuitive. In internal tests, the horizontal pressure gradient has been observed to increase dramatically between a turbine in isolation and a turbine at the first row of a farm. Could the authors explain this discrepancy? Could the authors confirm that the pressure gradient force for the front-row turbine in Figure 15a is normalized with the horizontal momentum advected only through the surface S_x of Figure 10c, instead of 10b? Please distinguish the variables for both S_x used.

We can confirm that the pressure gradient force is being normalized using the appropriate momentum flux at the inflow of the control volume. Given that there is no literature comparing the pressure gradient force for a wind plant and a single turbine, we cannot provide a definitive explanation for this discrepancy. When we evaluate the pressure gradient in the original numerical grid (without interpolation to the common dx = 15 m grid that is used in the control volume analysis), we clearly see that the pressure gradient upstream and downstream of a front-row turbine in the wind plant and a stand-alone turbine is almost perfectly overlapping (Figure vi).

[Figure]

*Figure vi: Pressure gradient in dimensional form for the moderate (top) and weak (bottom) stability cases within the turbine rotor layer.*

We add clarification in our analysis to highlight that this behavior is observed in our simulations specifically and propose this as an avenue of future research:

Line 284: "The pressure gradient upstream of a single front-row turbine in the wind plant is virtually the same as the pressure gradient force for a stand-alone turbine for both atmospheric conditions in our simulations (Figure 17)."

Line 404: "In addition, more research is needed to further validate the forcing mechanisms driving blockage for a front-row turbine in the wind plant and a stand-alone turbine for a wide range of atmospheric conditions."

We also modify Figure 11 and add clarification to its caption, and modify the captions of Figures 16,19 accordingly to distinguish between the different control volumes in our analysis.

[Figure]

Figure 11: Illustration of the region considered in the analysis of the momentum balance along the x-direction for the whole wind plant (a), a single turbine in the front row of the wind plant (c) and stand-alone turbine (d). The integral momentum equation is evaluated on differential control volumes $\delta V$ along the streamwise direction upstream of the turbines (b). Each control volume is bounded vertically by the top (z = 153 m) and bottom (z = 27 m) of the turbine rotor layer. Horizontally in the y-direction, the control volume spans the region upstream of the wind plant (from y = 1953 m to y = 5922 m). For the single and stand-alone turbine (c,d), the control volume is bounded in the y-direction by the rotor diameter. Each differential control volume is 15 m long in the x-direction. The area of each control surface $S_i$ is illustrated in the differential control volume $\delta V$ in Panel (b).

Figure 17,20: … In panel (a), the integral momentum equation is evaluated on differential control volumes as shown in Figure 11c for a single turbine in the middle of the wind plant and as shown in Figure 11d for a stand-alone turbine. In panel (b), the integral momentum equation is evaluated on the control volume V shown in Figures 11c,d for a single turbine in the middle of the wind plant and for a stand-alone turbine. The pressure gradient force is normalized using the u-momentum flux at the inflow of the control volume V in Figures 11c,d far upstream $\rho \overline{u}_\infty \overline{u}_\infty S_x$ for the respective stability case.

Furthermore, as the horizontal pressure gradient does not change across all the studied cases, the authors postulate that what drives the changes in the vertical momentum advection is a vertical pressure gradient developing upstream of the farm. This vertical pressure gradient seems then to be identified as the main mechanism causing global blockage. Unfortunately, most of the very little discussion on it is relegated to the Appendix. The authors should consider introducing the plots for the integral momentum balance in the vertical direction in the body of the paper and expand the discussion on the vertical pressure gradient.

Thank you for highlighting this point because it is important for us to clarify. We do not state that the changes in vertical transport of u-momentum are mainly caused by changes in the vertical pressure gradient. Rather, we identify that the differences in vertical advection of horizonal momentum between both atmospheric conditions are primarily due to differences in vertical shear of the vertical velocity. We added clarification in the manuscript as follows:

Line 297: "The vertical velocity advects horizontal momentum out of the turbine rotor layer (Figure 19). Vertical advection of horizontal momentum is 20% larger in the moderate stability case compared to the weak stability upstream of the first turbine row (Figure 19b). Larger vertical shear of the horizontal velocity in the moderate stability case compared to the weak stability case is the primary cause for the increased vertical advection of horizontal momentum. Shear $\left( \frac{\Delta \overline{u}}{\Delta z} = \frac{\overline{u}_t - \overline{u}_b}{D} \right)$ between the bottom $\overline{u}_b$ and top $\overline{u}_t$ of the turbine rotor layer is 43.6% larger in the -0.5 K/h simulation compared to the -0.2 K/h simulation. Similarly, the vertical velocity in the turbine rotor layer is 20% larger in the moderate stability case than in the weak stability case between x = -6D and x = 0D. The vertical velocity is expected to be larger in the moderate stability case because, as shown in Figure 18, the streamwise slowdown of the flow is transformed almost entirely into vertical motions. As a result, advection of horizontal momentum by the vertical velocity $\Delta(\rho \overline{u}\,\overline{w}\,S_z) = \rho S_z (\overline{u}_t \overline{w}_t - \overline{u}_b \overline{w}_b)$ is amplified."

We also replace "vertical momentum transport" with "vertical transport of horizontal momentum" throughout the text. As an example, line 309 now reads: "Vertical advection of horizontal momentum is amplified for a wind plant compared to a stand-alone turbine (Figure 20). For a

given atmospheric condition, vertical shear of the horizontal velocity remains unchanged between the stand-alone turbine and wind plant simulations. Therefore, differences in vertical transport of horizontal momentum between a stand-alone turbine and a turbine in the wind plant are entirely due to the vertical velocity that forms upstream of the turbine array (Figure 21)."

Further analysis should also be performed to make the point of the authors stronger. As done for the horizontal momentum, also the vertical momentum balance should be compared between the wind farm and the single wind turbine cases. The claim of the authors could be supported by demonstrating that the vertical pressure gradient increases in the wind farm case, in the same order as the blockage increases.

Please see comment above, where we clarify that the primary cause for an increase in vertical advection of horizonal momentum is larger shear of the vertical velocity $\left(\frac{\Delta \overline{u}}{\Delta z}\right)$ within the turbine rotor layer.

**On gravity waves:**

The authors write that gravity waves did not form in the weak free-atmosphere stratification simulation of Wu and Porté-Agel (2017) in Line 40-41. This statement might be misleading and it should be revised. In fact, Wu and Porté-Agel did not observe upstream propagating gravity waves in their simulation with a weak stratification in the free atmosphere. Wu and Porté-Agel (2017) differentiate between subcritical and supercritical flows. In subcritical flow gravity waves can move upstream, in supercritical flows they can't. Wu and Porté-Agel (2017) do not state that there are no gravity waves in case of the supercritical flow. My suggestion is to apply the theory presented in Wu and Porté-Agel (2017) in order to determine whether the cases shown by the authors are cases of supercritical flows.

Thank you for highlighting this inaccuracy, we fixed it in our revised manuscript. We also include the Froude number in our analysis and expand the discussion on gravity waves in Appendix C. We update the manuscript as follows:

Line 39: "Gravity waves propagate upstream in their strong free-atmosphere stratification simulation but do not propagate upstream in their weak free-atmosphere stratification simulation (Wu and Porté-Agel, 2017)."

Line 68: "Here, we investigate how atmospheric stability modifies upstream blockage with minimal upstream propagation of gravity waves (see Appendix C for a discussion on gravity waves in our domain)."

Line 490: "We also evaluate the upstream propagation of gravity weaves in our simulations using the Froude number, as done by Wu and Porté-Agel (2017). The Froude number characterizes the balance between flow acceleration or deceleration and the pressure gradient imposed by the displacement of the stably stratified flow $Fr = \overline{U}/\sqrt{g'H}$, where $\overline{U}$ is the boundary-layer bulk velocity, $g' = \frac{g\Delta\theta}{\theta_0}$ is the reduced gravity accounting for the inversion strength, and H is the boundary layer height. Wu and Porté-Agel (2017) suggest gravity waves amplify the blockage effect in subcritical flow (Fr < 1), where pressure disturbances propagate upstream. The Froude number in our weak and moderate stability simulations is 1.2 and 1.35,

respectively, characteristic of supercritical flow (Fr > 1), thus pressure disturbances do not propagate upstream."

**Line 43-45:** "Note that Allaerts and Meyers (2017, 2018); Maas (2022) simulate the flow around an infinitely wide wind plant; therefore, the large vertical boundary layer displacement that excites gravity waves (and thus the velocity deceleration in the induction region) is likely overestimated compared to operational wind plants." This statement is not obvious. Please add a better explanation based on existing literature here.

We include the following for clarification:

Line 42: "Note that Allaerts and Meyers (2017, 2018); Maas (2022) simulate the flow around an infinitely wide wind plant. The power loss due to upstream-propagating gravity waves increases as the wind plant becomes infinitely wide (Allaerts and Meyers, 2019). Therefore, the velocity deceleration in the induction region of an infinitely wide wind plant is likely larger than would be expected in an operational wind plant of finite width."

**On the choice of grid spacing (Line 89-91):**

Did the authors carry out any sensitivity tests in order to show that the grid spacing used by them is actually sufficiently fine? If not this should be mentioned in the manuscript.

Because another reviewer also asked about grid resolution, we now include an additional appendix in our manuscript:

Line 494: Appendix D

"Grid resolution in our simulations is sufficient to resolve most turbulence kinetic energy across the turbine rotor layer (Figure D1). For the non-linear backscatter and anisotropy subgrid-scale turbulence model (Kosović, 1997), the total turbulence kinetic energy $\overline{k}_{tot}$ is given as $\overline{k}_{tot} = \frac{1}{2}\left(\overline{u_i'u_i'} + m_{ii}\right) + \overline{k}_{SGS}$, where $\overline{u_i'u_i'}$ represents the resolved TKE, $m_{ii}$ are the normal subgrid-scale stress components, and $\overline{k}_{SGS}$, is the subgrid-scale TKE. Nearly 80% of TKE in the turbine rotor layer is resolved by the numerical grid for both simulations (Figure D1). Because less than 80% of TKE in the lower rotor layer is resolved in the weak stability case ($\overline{k}_{res}/\overline{k}_{tot}$= 0.78 at z = 30 m), a finer grid is used for the simulation of moderately stably stratified flow."

[Figure]

**On the choice of the model domain (Figure 1):**

Did the authors check whether the 45 D long part of the model domain upstream of the wind farm is actually sufficiently long enough in order to avoid that the inflow boundary has an impact on the blockage that is found in the simulations?

We investigate the effect from prescribed boundary conditions by considering the velocity field close to the domain boundaries. We find that boundary conditions have minimal effects on the flow in both the induction and wake regions. Because the velocity deceleration in the induction region is virtually equal to freestream 30D upstream of the turbines (see Figure 8 below), we conclude that a 45D fetch is adequate to minimize the effects of the upstream boundary condition. At the downstream end of the domain, the wake recovery only appears to be influenced by the boundary conditions very close to the domain boundary.

We add clarification to the text as follows:

Line 107: "As will be shown later in the manuscript, the velocity deceleration in the induction region is virtually zero 30D upstream of the wind plant. Therefore, 45D of fetch upstream of the wind plant is deemed sufficient to investigate the induction region of the turbine array."

Furthermore, we adjust the x-axis in Figure 8 to show the velocity deceleration far upstream as follows:

Line 183: "The velocity deceleration asymptotes to zero far upstream (x<-30D) for both atmospheric conditions (Figure 8)."

[Figure]

Figure 8: Normalized velocity deficit $\left( \Delta U = \frac{\overline{U}-\overline{U}_\infty}{\overline{U}_\infty} \right)$ for the inter- and intra-turbine regions upstream of the wind plant for each atmospheric condition…

Did the authors check whether the space in y-direction at the side of the the wind farm is sufficiently large in order to be able to exclude that the simulation results are disturbed by the lateral boundaries? Does the simulation approach the case of an isolated wind farm or that of an infinite wind farm in y-direction?

We acknowledge that the domain size in the streamwise direction can affect blockage. However, we did not perform a sensitivity analysis on the space to the sides of the wind plant because of its high computational costs. We add the following information to the manuscript:

Line 110: "Strickland and Stevens (2022) show the power of front-row turbines in a wind plant is sensitive to the ratio between the wind plant width ($L_{y-wp}$) and the domain size in the y-direction ($L_y$). Because the change in turbine power for ratios $L_{y-wp}/L_y$< 0.5 is small (Strickland and Stevens, 2022) but the increase in computational resources is significant, we use a ratio of 0.5 here."

**On the set-up of the large-eddy simulations (Section 2.1):**

As the geostrophic wind is used as a boundary condition in the simulations, an information on the geographic coordinates where the simulations are carried out should be provided. The geographic coordinates will change the Coriolis parameter and therefore also the profiles of the wind components.

Thank you for pointing this out. We include this information as follows:

Line 127: "Both simulations are initialized with a $U_g = 11$ m/s geostrophic wind speed and the Coriolis parameter is $f_c \approx 9.37 \times 10^{-5}$ 1/s, corresponding to a latitude of 40°."

**On Figure 2:**

When is the averaging period that is used in this Figure? Has the simulation reached a stationary state yet?

We add clarification in the caption as follows:

Figure 6: Horizontal wind speed (a), wind direction (b) and potential temperature (c) profiles for the atmospheric conditions simulated herein. Atmospheric variables are averaged spatially over the entire domain and temporally over 1 h after spin-up of turbulence and stability is complete.

**On Figure 3:**

A line illustrating a function of the type y=A+k^(-5/3) (Kolmogorov slope) should be added in order to show that the simulation actually resolves a part of the inertial subrange of turbulence.

We now include this information in the figure. Also, as requested in comments below, we make the colormap consistent in each panel.

[Figure]

Figure 2: Compensated turbulence spectra of the w-velocity for the $\Delta x = 7m$ neutrally stratified boundary layer in the precursor simulation at z=90 m (a), z=300 m (b), and z=800 m (c). Colored lines indicate time since initialization in 20-minute time increments. The dotted, black vertical line in each plot represents the effective grid resolution (4-5 $\Delta x$) expected from the reduced advection scheme in our simulations (Kosović et al., 2016). The theoretical −2/3 Kolmogorov slope for the inertial range is indicated by the solid black line in each plot.

**On Figure 5:**

The Figure shows clearly an inertial oscillation that is triggered when the cooling of the atmospheric boundary layer starts. Obviously, the inertial oscillation has not been completely damped after 8 h. What does this mean for the analysis of the global blockage effect?

We acknowledge that the stable boundary layer is still evolving, and this is expected because of the cooling rate at the surface. However, the effect from this boundary layer evolution is minimal over the simulation time (1 h). As shown in the shaded areas in Figure vii, the change in hub-height wind direction, a proxy for the inertial oscillation, over the simulation period (1 h) is smaller than 1 degree.

[Figure]

*Figure vii: Time series of hub-height wind direction for each atmospheric condition over the simulation period. The shaded area in each plot represents the simulation time period for evaluating blockage.*

We include the following in the manuscript:

Line 168: "Over the simulation period, hub-height wind direction varies by less than 1° for both atmospheric conditions."

We also direct the reviewer to the discussion of the physical mechanisms modifying blockage where we discuss the effect of Coriolis forcing, which drives the inertial oscillation:

Line 212: "Even though the Coriolis force in our simulation domain is not negligible, Coriolis forcing in the induction region is small. The Coriolis parameter scales as $f_c \sim 10^{-4}\ s^{-1}$ and the v-velocity in the turbine rotor layer for both stability cases is on the order of $\bar{v} \sim 0.1\ m/s$, thus Coriolis forcing is of the order $f_c \bar{v} \sim 10^{-5}\ m/s^2$."

**On Figure 6:**

It is difficult to show with Figure 6 that the Monin-Obukhov-length and the Richardson number have already reached stationary values within the simulation time used (even for the case with the stronger cooling rate).

We don't expect the stability metrics in our simulations to reach a steady state because we have a cooling rate at the surface, which is anticipated to result in a continual evolution of the boundary layer. We provide clarification in the text as follows:

Line 155: "The -0.5 K/h simulation is run until the temporal change in bulk Richardson number in the surface layer and the Obukhov length is small."

Line 162: "Note that we do not expect our simulations to reach a steady state because the cooling rate at the surface continually modifies stability in the surface layer. Nonetheless, the evolution of the surface layer is slow after 8 h (13 h) for the moderate (weak) stability simulation."

**Line 201:**

When reading the manuscript I understood that independent of the simulation each differential control volume has an extension of 15 m along the x-direction. However, how does this work out when the grid spacing is 7 m in one of the two cases simulated?

Thank you for catching this, we interpolate both numerical grids to a common grid with horizontal spacing equal to 15 m, which is close to a common multiple of both grid resolutions. We tested different common grids, but there were no appreciable differences in the results. We clarify this in the manuscript:

Line 222: "Because grid spacing is different for the stability cases, we interpolate atmospheric variables from each simulation to a common grid with horizontal resolution of 15 m."

**Figure 11:**

The authors should elaborate a bit more on the explanation of the observation that the decrease of pressure starts already slightly upstream of the actuator disk.

The maximum in the pressure perturbation field upstream of the turbine is located slightly upstream of the GAD model. The maximum in the pressure field upstream of the turbines has been reported in the literature (e.g., Figure 12 in Strickland and Stevens, 2022).

We add the following Figure and clarification to the text:

Line 235: "The thrust force imparted by the turbine to the flow is a fundamental driver for blockage (Ebenhoch et al., 2017). In the numerical implementation of the GAD model, the aerodynamic forces are spread across multiple grid cells along the streamwise direction to avoid numerical instabilities (Mirocha et al., 2014). A pressure gradient upstream of the turbine forms in response to the thrust force that the turbine imparts on the flow ($\Delta\overline{p}_{pert}/\Delta x > 0$ upstream of the turbine in Figure 12). Because the thrust force is spread across multiple grid cells in the streamwise direction, the maximum in pressure in front of the turbines is located slightly upstream of the actual location of the GAD in the numerical domain (Figure 12). As a result, we restrict the control volume V in Figure 11 to extend up to x=5647 m, the location of the maximum in pressure perturbation upstream of the turbine array (vertical dotted line in Figure 12)."

[Figure]

Figure 12: Hub-height pressure perturbation of a front-row turbine in the wind plant for each stability case. The pressure perturbation is normalized over the corresponding dynamic pressure for each stability condition. The solid black vertical line illustrates the location of the GAD in the numerical domain. The dotted vertical line illustrates the local maximum in pressure perturbation upstream of a front-row turbine in the wind plant. The secondary x-axis is scaled to locate x=0D at the location of the front-row turbine.

Line 246: "Immediately upstream of the turbine (cross-hatched area in Figure 13), the pressure gradient force becomes negative because the GAD produces a pressure drop in the flow and the pressure perturbation field reaches a local maximum slightly upstream of the turbine (Figure 12). In the numerical implementation of the GAD model, the aerodynamic forces are spread across multiple grid cells to avoid numerical instabilities (Mirocha et al., 2014), which causes the pressure field to decrease over multiple grid cells (Figure 12)."

We also modify the relevant figures in the manuscript (Figures 13, 14, 16, 17, 18, 19, 20) to highlight this pressure drop that happens across the grid cells that intersect the GAD. An example is Figure 13 in the manuscript:

[Figure]

Figure 13: Streamwise evolution of the u-momentum equation (Eq. 2) for a stand-alone turbine in the weak (a) and moderate (b) stability cases. The integral momentum equation is evaluated on differential control volumes $\delta V$ along the x-direction, as shown in Figure 11. The x-axis is scaled to locate x=0D at the location of the turbine. The mean momentum fluxes and the pressure gradient force are normalized using the u-momentum flux at the inflow of the control volume far upstream ($\rho \overline{u}_\infty \overline{u}_\infty S_x$) for the respective stability case. The cross-hatched area in each panel illustrates the grid cells influenced by the thrust force from the GAD.

We also add clarification in the text as follows:

Line 255: "Note that the momentum balance immediately upstream of the turbine (cross-hatched area in Figure 13) is not equal to zero because the thrust force from the GAD is not included in our calculations."

**Figure B1:**

This is one of the main findings and should be integrated in the results part of the manuscript.

Please see comment above, where we clarify that the primary cause for an increase in vertical advection of u-momentum is larger shear of the vertical velocity $\left(\frac{\Delta \overline{u}}{\Delta z}\right)$ within the turbine rotor layer.

**General and technical comments**

- Whole document: Use non-italic units, introduce a Space between number and unit, avoid line breaks between number and unit, add clickable links in the pdf for references on figures etc.

We appreciate this suggestion. We use non-italic units throughout the text, include a space in between the number and unit, and add clickable links throughout the pdf.

- Line 17: Please add a reference for wind turbine and cluster wakes each.

We update the text as follows:

Line 19: "Wind turbine and wind plant wakes can also affect power production of downstream turbines (El-Asha et al., 2017) and plants (Stieren and Stevens, 2022), an effect known as wake loss."

- Line 18-19: Why are so many references given here? One reference with a more general view on blockage like Bleeg et al., 2018 is sufficient, the others will be addressed in the state of the art.

Thank you for your suggestion, we modify the manuscript to only include reference to Bleeg et. al (2018).

- Line 21: "know" needs to be replaced by "known"

We modify the manuscript accordingly.

- Line 29-30: Please sort relevant references to the named factors influencing global blockage (" size and layout of the wind plant, atmospheric conditions, wind turbine characteristics, and wind speed")

As requested by the reviewer, we sort the references as follows:

Line 27: "The velocity deceleration within the induction region can vary substantially depending on the size and layout of the wind plant (e.g., Centurelli et al., 2021; Strickland and Stevens, 2022; Strickland et al., 2022; Bleeg et al., 2018), atmospheric conditions (e.g., Allaerts and Meyers, 2018, 2017; Bleeg and Montavon, 2022; Schneemann et al., 2020; Strickland et al., 2022), wind turbine characteristics (e.g., Ebenhoch et al., 2017), and wind speed (e.g., Schneemann et al., 2020)."

- Line 33-35: The named references show different velocity deficits in different distances upstream. Please specify the general statement here.

Thank you for your suggestion. We considered including a more detailed description of the velocity deceleration from each study. However, the objective of this paragraph is to highlight the order-of-magnitude difference in the velocity deficit reported in some studies, rather than the velocity deceleration from each research paper. Therefore, we consider that providing a general statement is more valuable to the reader than giving detailed information about each reference.

- Line 62: "neutral LES" This is uncommon terminology. Please change to e.g. "The authors simulated a neutrally stratified boundary layer flow using LES..."

Line 63 now reads: "Using LES of neutrally stratified boundary-layer flow, Strickland and Stevens (2022) show an increase in the adverse pressure gradient upstream of wind plants with closely-spaced turbines in the cross-stream direction."

We also update the captions of Figures 2 and 3 to replace "neutral LES" with "neutrally stratified boundary layer".

- Line 89: The terminology introduced here for the state of the atmosphere should be kept throughout the document. The authors simulate a weakly stable boundary layer and a moderately stable boundary layer. Changing this to "moderately and weakly stratified flow" without referring to "stable" is misleading.

Thank you for your suggestion. We update the manuscript so that we always refer to stably stratified flow, following the convention found in the literature (e.g., Allaerts and Meyers, 2018). For example, Line 349 now reads: "The wind speed is 3.5% and 2.8% slower than freestream 2D upstream of the wind plant for moderately and weakly stably stratified flow."

- Figure 1: Please add the turbine spacing in x and y direction in the Figure or the caption.

We update the caption in Figure 1 as follows:

Figure 1: Relative location of the turbines in the wind plant (a) and stand-alone turbine (b) simulations for evaluating blockage. Forty NREL 5MW wind turbines constitute the 200MW wind plant simulated herein. Turbine spacing is 7D and 3.5D in the streamwise and cross-stream directions, respectively.

- Table 1: Please add information about the stability, i.e. the cooling rate applied.

We updated Table 1 in the manuscript to include the cooling rate at the surface for each stability condition.

- Line 113: Adding the word "temporal" makes the method more clear here: ... we prescribe a temporal cooling rate rather than...

We add the word "temporal" as suggested. Line 121 now reads: "… we prescribe a temporal cooling rate rather than a heat flux at the surface."

- Line 113: flow -> flows are...

We modify the manuscript accordingly.

- Figure 2: Please label the height axis as z. Please add information about the period the data is averaged on. How long was the cooling applied before the averaging period? Further, Figure 2 shows the resulting profiles while Figure 3 and 4 jump back to the pre run simulations. This is a bit confusing while reading and should be restructured.

Thank you for your suggestions. We replace the "Height" label with "z" throughout the manuscript (Figures 3, 4, 6, A2, A4, D1). We move the figure showing the final atmospheric conditions to the end of section 2.2 so that it appears after we describe spin-up for our simulations. We also update the caption to include information about the temporal averaging as follows:

Line 165: "Atmospheric conditions after spin-up of turbulence and stability are shown in Figure 6."

Figure 6: Horizontal wind speed (a), wind direction and potential temperature (c) profiles for the atmospheric conditions simulated herein. Atmospheric variables are averaged spatially over the entire domain and temporally over 1 h after spin-up is complete.

- Figure 3: The legend just holds two entries while the Figure includes many different curves / colors. The colours seem not to be consistent through the subplots. The evolution is hard to

follow and the description from Line 126 ff could not be reproduced. Please change the legend or introduce a colour scale. In my opinion the amount of curves shown can and should be reduced. Further, please add the Kolmogorov slope into the plots. Please state in the caption that this is a plot of a pre-run of the LES.

We appreciate your suggestions to make this figure clearer. We modify the figure so that the colormap is consistent across the different panels. We also reduce the number of curves shown in half and include the theoretical Kolmogorov slope. We update the caption to clarify that these results are for the precursor neutral LES.

Figure 2: Compensated turbulence spectra of the w-velocity for the $\Delta x = 7$m neutrally stratified boundary layer in the precursor simulation at z=90 m (a), z=300 m (b), and z=800 m (c). Colored lines indicate time since initialization in 20-minute time increments. The dotted, black vertical line in each plot represents the effective grid resolution (4-5 $\Delta x$) expected from the reduced advection scheme in our simulations (Kosović et al., 2016). The theoretical −2/3 Kolmogorov slope for the inertial range is indicated by the solid black line in each plot.

- Figure 4: z for height axis, readable legend covering all cases as for Figure 3. Maybe Figure 3 and 4 can be even combined.

We incorporated your suggestions in the text. As described above, we changed the label for the y-axis to "z". Also, like in the previous comment, we reduce the number of curves and update the caption to clarify that these results are for the precursor neutral LES and that the colored lines correspond to different times since the simulation is initialized. We decided not to combine Figures 2 and 3 because one relates to turbulence development and the other to the mean flow.

Figure 3: Vertical profile of the horizontal wind speed for the $\Delta x = 7$ m neutrally stratified boundary layer. The velocity profile is averaged spatially over the entire domain. Colored lines indicate time since initialization in 20-minute time increments.

- Caption Figure 4: "Horizontal velocity profile" is misleading, e.g. "vertical profile of the horizontal wind" is more clear.

We clarify the caption of Figure 3 as shown in the comment above.

- Line 131: Delete "On average", the information is double.

We modify the manuscript accordingly.

- Line 149: "Nose" is a not common expression for the wind speed maximum of a LLJ. Please use a more common expression.

We appreciate your suggestion, however the atmospheric science literature commonly refers to wind speed maximum of the low-level jet as the nose (e.g., Banta et al., 2002; Vanderwende et al., 2015; Carroll et al., 2019; Brogno et al., 2021; Smith et al., 2019).

- Figure 6: Where is the benefit in showing both L and Ri_bulk? Further, both cases seem not to have converged. Please elaborate on this.

As we mention in the comments above, we don't expect the stability metrics in our simulations to reach a steady state because we have a cooling rate at the surface, which is expected to

result in a continual evolution of the boundary layer. We see value in showing both stability parameters because L describes stability at the surface whereas Ri describes stability across the turbine rotor layer, both of which are important in our simulations. We provide clarification in the text as follows:

Line 155: "The -0.5 K/h simulation is run until the temporal change in bulk Richardson number in the surface layer and the Obukhov length is small."

Line 162: "Note that we do not expect our simulations to reach a steady state because the cooling rate at the surface continually modifies stability in the surface layer. Nonetheless, the evolution of the surface layer is slow after 8 h (13 h) for the moderate (weak) stability simulation."

- Line 155: Better use ° instead of deg in the whole document.

Thank you for your suggestion, we modify the manuscript and figures accordingly.

- Figure 7: A second x-axis in units of D could support readability. Within the stippled areas the wind field cannot be well seen. We suggest to remove the stipples and just keep the bordering lines and lables. Why is the wake region marked as well?

Thank you for your suggestion. We considered removing the stippled areas; however, the only reason for this figure is to clearly define the inter- and intra-turbine regions, and the stippled areas are very effective in doing so. Therefore, we decide to leave them in the figure. Nonetheless, we update the figure so that the stippled areas only cover the induction region of the turbines and not the wakes.

- Figure 9: Caption could refer to Figure 8 saving copy/pasted information. Even better could be to combine both plots as subplots in a single Figure. Same could be applied for some of the following results plots.

As pointed out in the Community Comment (CC1), we had a typo on the caption. We updated the caption to emphasize that the velocity deficit is averaged in the y-direction over the intra-turbine region, which corresponds to the stippled area in Figure 7.

- Line 192, Equation 1: Labelling of the different terms can help the reader to follow more easily.

Thank you for your suggestion, we include labelling in Equation 1 and Equation B1.

- Line 224ff: The small differences described could not be seen in the Figure. At which position does the difference occur?

We clarify as follows:

Line 259: "In the entire control volume V in Figure 11d, the pressure gradient force that drives flow deceleration upstream of the turbine differs by 3.1% between atmospheric conditions."

- Line 254: "atmoshperic" needs to be changed to atmospheric

We modify the manuscript accordingly.

- Line 259: Please elaborate a bit more on the mentioned secondary circulation as it is not obvious to the reader.

We find that the change in u-velocity is transformed into a change in w-velocity. Also, as requested by another reviewer, we clarify the manuscript and replace "secondary circulation" with "secondary flow feature" as follows:

Line 293: "Mass balance indicates the slowdown of the u-velocity in the turbine rotor layer $(\Delta(\rho \overline{u} S_x) < 0)$ is balanced by the development of a secondary flow feature in the form of net-upwards vertical motion $(\Delta(\rho \overline{w} S_z) > 0)$ for both stability conditions (i.e., $\Delta(\rho \overline{u} S_x) + \Delta(\rho \overline{w} S_z) \approx 0$)."

- Line 260: " The increase in vertical velocity is driven by a vertical pressure gradient..." What is the driver for this pressure gradient? This seems to be one of the most relevant findings, please better explain and highlight.

We direct the reviewer to the comments above, where we show that the primary amplifier for blockage is vertical shear of the horizontal velocity over the turbine rotor layer. We also want to stress that the idea of the vertical momentum analysis is to show that vertical motions can form in stably stratified flow, even though there is a downward buoyancy force. We clarify as follows:

Line 296: "The development of the vertical velocity is possible because of a vertical pressure gradient that balances the downward buoyancy force in the stably stratified flow (see Appendix B for a deeper analysis on vertical momentum balance)."

- Line 287: please change "x1.9" to 1.9 times

We modify the manuscript accordingly.

- Line 291-292: Bleeg et al. (2018) suggest (plural)

We modify the manuscript accordingly.

- Line 332-343: Please elaborate more on the difference between momentum advection and a deflection of momentum upwards. This is not obvious.

We direct the reviewer to Line 341 in the revised manuscript, where we elaborate on momentum advection and flow deflection as follows:

Line 344: "The slowdown of the u-velocity in the induction region of the wind plant is transferred into vertical motions (Figure 18). Other simulation studies have also noted this vertical deflection of the flow (e.g., Wu and Porté-Agel, 2017; Allaerts and Meyers, 2017). The vertical velocity advects horizontal momentum out of the turbine rotor layer."

We also add clarification as follows:

Line 382: "… they suggest an increased pressure gradient amplifies blockage as cold air is deflected upwards (i.e., u-velocity is transformed into w-velocity)."

**Literature**

Allaerts, D. and Meyers, J.: Boundary-layer development and gravity waves in conventionally neutral wind farms, Journal of Fluid Mechanics, 814, 95–130, https://doi.org/10.1017/jfm.2017.11, 2017.

Allaerts, D. and Meyers, J.: Gravity Waves and Wind-Farm Efficiency in Neutral and Stable Conditions, Boundary-Layer Meteorology, 166, 269–299, https://doi.org/10.1007/s10546-017-0307-5, 2018.

Bleeg, J., Purcell, M., Ruisi, R., and Traiger, E.: Wind Farm Blockage and the Consequences of Neglecting Its Impact on Energy Production, Energies, 11, 1609, https://doi.org/10.3390/en11061609, 2018.

Maas, O.: From gigawatt to multi-gigawatt wind farms: wake effects, energy budgets and inertial gravity waves investigated by large-eddy simulations, https://doi.org/10.5194/wes-2022-63, 2022.

Schneemann, J., Theuer, F., Rott, A., Dörenkämper, M., and Kühn, M.: Offshore wind farm global blockage measured with scanning lidar, Wind Energy Science, 6, 521–538, https://doi.org/10.5194/wes-6-521-2021, 2021.

Strickland, J. M. and Stevens, R. J.: Investigating wind farm blockage in a neutral boundary layer using large-eddy simulations, European Journal of Mechanics - B/Fluids, 95, 303–314, https://doi.org/10.1016/j.euromechflu.2022.05.004, 2022.

Wu, K. and Porté-Agel, F.: Flow Adjustment Inside and Around Large Finite-Size Wind Farms, Energies, 10, 2164, https://doi.org/10.3390/en10122164, 2017.

**Response to Referee #3: James Bleeg**

Authors' responses to reviewer comments appear in blue text. Line numbers referenced in the authors' responses refer to the revised document. Figures with Arabic numerals (e.g., Figure 10) correspond to the revised manuscript, Figures with roman numerals (e.g., Figure iv) only appear on the response to reviewer's comments.

**Paper title:** "Investigating the physical mechanism that modify wind plant blockage in stable boundary layers"

The authors simulated an idealized wind farm on flat terrain operating in two different stable boundary layers. The simulations were run back-to-back with a turbine operating in isolation. The results of these simulations were then analyzed to quantify the impact of blockage on the flow upstream of the wind farm and also on the upstream row of turbines. Further, the authors interrogated the solutions to understand the physical mechanisms behind these blockage effects.

If accepted for publication, I think this paper would be a strong addition to the growing literature on wind farm blockage effects. The analysis of the simulated flow is lucid and compelling. The main findings, in my view, are important and mostly new.

The main area where I feel the paper could be improved is putting the research into context. The next section of this review focuses on this aspect. I include a number of miscellaneous comments after the context section.

**Putting the findings into context:**

**Gravity waves** – Gravity waves are referred to a number of times in the manuscript, mostly in the vein of we-are-not-dealing-with-gravity-waves-in-this-study. The introduction puts this explicitly: "Here, we investigate how atmospheric stability modifies upstream blockage in the absence of gravity waves." There is even a section in the appendix substantiating the claim that there are no gravity waves in the flow solutions. I do not think this approach to setting the no-gravity-wave stage is necessary and may even be counterproductive, potentially leading readers to believe that the findings are limited to gravity-wave free conditions.

There are good reasons to believe wind-farm-induced gravity waves are very common. The manuscript implies the findings in the two Bleeg papers are not affected by gravity waves (see line 50), but this is not the case. All our wind farm simulations have gravity waves, at least to some degree. Gravity waves, as you know, are the primary means by which disturbances are transmitted in density-stratified flow. Such a disturbance might be an obstacle like a wind farm causing flow to rise. The resulting gravity waves—or speaking more generally, inviscid effects related to disturbed stratified flow—and their effect on the wind farm will depend upon the characteristics of the stratification and the wind farm itself. The authors appear to be simulating a set of conditions where the effects of gravity waves on flow upstream of the wind farm is not as pronounced as in the first Allaerts and Meyers JFM paper. I think it is sufficient to say just that.

There is strong numerical evidence at this point that inviscid effects related to stratification above the boundary layer—and the horizontal pressure gradients associated with them—can

have a significant impact on blockage effects. This paper helps bring to light another important influence on the production of leading row turbines relative to what they would produce in isolation: the vertical advection of u-momentum. The finding not only improves our physical understanding of blockage effects, but also has significant implications for the modelling of these effects.

We appreciate and share your opinion on the prominent discussion on gravity waves. However, other reviewers have strong opinions about spurious gravity waves in the domain and upstream propagation of gravity waves, which can amplify blockage. Therefore, we decide to leave this discussion in our manuscript to proactively address such concerns. However, we do soften our language throughout the entire manuscript. An example is as follows:

Line 68: "Here, we investigate how atmospheric stability modifies upstream blockage with minimal upstream propagation of gravity waves"

**Engineering models for blockage –** The engineering models designed to predict blockage effects are inviscid, potential flow-type models, which do not account for shear and therefore are not able to reliably account for what this paper suggests is one of the most important contributors to front row blockage loss: the vertical advection of u-momentum. It is up to the authors, but I think it might be worth emphasizing this in the paper.

The manuscript states that blockage is not currently accounted for in EYA's. This was true in 2018, but it is not true anymore. Almost all EYA's account for blockage in one way or another. And in many cases, a potential flow model is used to account for the blockage effect. Potential flow models, as discussed, will miss a significant contributor to the impact of blockage on turbine production. I feel that the wind industry community would benefit from having this point highlighted.

Thank you for your suggestion, this is indeed worth highlighting. We include the following in our discussion:

Line 375: "Potential flow models often used to include the power loss from blockage in energy assessments do not account for vertical shear of the horizontal velocity (e.g., Forsting et al., 2021). Vertical advection of u-momentum is the primary amplifier for blockage in our simulations, driven by shear. As a result, energy yield estimates might underestimate losses from blockage."

**Has anyone looked into this before?** According to the manuscript, the dominant mechanism causing front row wind turbines to produce less than they would in isolation is the vertical advection of u- momentum (at least in these simulations). To me, this is the most important finding in the paper. The Discussion and conclusion sections highlights others who have focused on adverse pressure gradients as a key driver behind blockage, and then explains that vertical advection of u-momentum amplifies the impact on the front row turbines. I'm a bit biased here, but it is worth mentioning that Bleeg and Montavon included a full section on this subject? The section makes the point that the combination of shear and flow rising as it approaches the wind farm, due to the presence of the wind farm and the ground, "appears to be an important factor in determine the magnitude of the blockage loss." Immediately following, the paper reads, "in case 9, for example, the streamline passing through the hub in the wind farm configuration originates approximately 4.5 m below the hub-intersecting streamline in the isolated case. In turn, the wind speed on the streamline far upstream is approximately 2% lower in the wind farm

configuration compared to isolated operation. This significant wind speed difference is the result of the vertical flow deflection combining with the increased shear that prevails in stable conditions."

In my opinion, the physical explanation provided in the manuscript under review is more clear, complete, and convincing that what we provide in Bleeg and Montavon. Much credit is due to the authors for this important finding. That said, I think it is fair to say that the Bleeg and Montavon paper did highlight the important influence of shear in combination with the upstream vertical deflection of flow as it relates to the impact of blockage on the production of leading row turbines relative to a turbine in isolation.

I am not an academic and am not familiar with what is required in a situation like this (also, I suppose I am not without bias in this regard), so I leave it to the authors and the editor to decide whether this should be acknowledged in the paper. For what it is worth, I think referencing the earlier result could further strengthen the credibility of the current finding. A sentence or two would be sufficient. And if it is followed by something indicating that your analysis more complete, I would not object, because it is.

Thank you for pointing out that we do not include this finding in our discussion. We agree that we should include previous work highlighting the importance of shear. We update the manuscript as follows:

Line 372: "Bleeg and Montavon (2022) also highlight the importance of vertical shear of the horizontal velocity and the vertical deflection of the flow. They suggest that, due to shear, the hub-height flow at the turbine location is slower than far upstream because the flow is being deflected upwards (Bleeg and Montavon, 2022)."

**Miscellaneous:**

The following questions, comments, and suggestions are in roughly order of priority:

    A.  I can't quite tell where the overall control volume in Figure 13 ends in the x-direction. The caption says it is bounded at the first turbine row (x = 5670 m). Clear enough, but where is that location in Figure 12? Is it at 0 D or just upstream? The reason why I ask is that that thrust body forces from the GAD are being applied to cells that include locations just upstream of 0 D, resulting in a rapid drop in pressure. If the end of the control volume in Figure 13 does correspond to 0 D, how would the results change if the end of the control volume were moved to just in front of the GAD?

Thanks for highlighting this inaccuracy. As you point out, the GAD applies forces to the flow upstream of its location due to its numerical implementation, resulting in a very rapid drop in pressure. We considered this in our initial analysis and shifted the control volume slightly upstream. We clarify as follows:

Line 235: "The thrust force imparted by the turbine to the flow is a fundamental driver for blockage (Ebenhoch et al., 2017). In the numerical implementation of the GAD model, the aerodynamic forces are spread across multiple grid cells along the streamwise direction to avoid numerical instabilities (Mirocha et al., 2014). A pressure gradient upstream of the turbine forms in response to the thrust force that the turbine imparts on the flow ($\Delta \bar{p}_{pert}/\Delta x > 0$ upstream of the turbine in Figure 12). Because the thrust force is spread across multiple grid cells in the

streamwise direction, the maximum in pressure in front of the turbines is located slightly upstream of the actual location of the GAD in the numerical domain (Figure 12). As a result, we restrict the control volume V in Figure 11 to extend up to x=5647 m, the location of the maximum in pressure perturbation upstream of the turbine array (vertical dotted line in Figure 12)."

[Figure]

Figure 12: Hub-height pressure perturbation of a front-row turbine in the wind plant for each stability case. The pressure perturbation is normalized over the corresponding dynamic pressure for each stability condition. The solid black vertical line illustrates the location of the GAD in the numerical domain. The dotted vertical line illustrates the local maximum in pressure perturbation upstream of a front-row turbine in the wind plant. The secondary x-axis is scaled to locate x=0D at the location of the front-row turbine.

Figure 15: Momentum balance over the entire induction region of the wind plant. The integral momentum equation is evaluated on the control volume V shown in Figure 11a. The control volume V is bounded in the x-direction by the inflow of the domain and the maximum in pressure perturbation upstream of the turbines (x = 5647 m). The mean momentum fluxes and the pressure gradient force are normalized using the u-momentum flux at the inflow of the control volume far upstream ($\rho \overline{u}_\infty \overline{u}_\infty S_x$) for the respective stability case.

We also modify the relevant figures in the manuscript (Figures 13, 14, 16, 17, 18, 19, 20) to highlight this pressure drop that happens across the grid cells that intersect the GAD. An example is Figure 13 in the manuscript:

Line 246: "Immediately upstream of the turbine (cross-hatched area in Figure 13), the pressure gradient force becomes negative because the GAD produces a pressure drop in the flow and the pressure perturbation field reaches a local maximum slightly upstream of the turbine (Figure 12). In the numerical implementation of the GAD model, the aerodynamic forces are spread across multiple grid cells to avoid numerical instabilities (Mirocha et al., 2014), which causes the pressure field to decrease over multiple grid cells (Figure 12)."

[Figure]

Figure 13: Streamwise evolution of the u-momentum equation (Eq. 2) for a stand-alone turbine in the weak (a) and moderate (b) stability cases. The integral momentum equation is evaluated on differential control volumes $\delta V$ along the x-direction, as shown in Figure 11. The x-axis is scaled to locate x=0D at the location of the turbine. The mean momentum fluxes and the pressure gradient force are normalized using the u-momentum flux at the inflow of the control volume far upstream ( $\rho \overline{u}_\infty \overline{u}_\infty S_x$) for the respective stability case. The cross-hatched area in each panel illustrates the grid cells influenced by the thrust force from the GAD.

Line 255: "Note that the momentum balance immediately upstream of the turbine (cross-hatched area in Figure 13) is not equal to zero because the thrust force from the GAD is not included in our calculations."

B. Again, I found the analysis of the results in sections 3 and 4 to be clear and convincing. That said I wonder if you could go just a little further to provide more physical insight and help the reader connect the dots towards your key finding. I refer specifically to the significant influence of the vertical advection of u-momentum. Perhaps you could break this down a bit. It could help drive home the point of the importance of shear. The x-component of velocity is clearly higher at the top of the control volume than the bottom. I'm not sure how the vertical component of velocity varies streamwise upstream of the wind farm, but I suspect it is positive and generally increases as flow approaches the wind farm. Of course, the trend may differ between the top and bottom of the control volume (in fact, the vertical component of velocity may be negative close to the rotors). I think it would be nice to have these things related to the vertical advection of u-momentum broken down, though I concede that what you already have in the report is sufficient to make your point.

We appreciate your suggestion. We now include a new figure of the vertical velocity in the manuscript and extend the discussion on shear. Note that we only show the vertical velocity for the upper half of the turbine rotor layer because, as you point out, the vertical velocity can be negative at the bottom of the turbine rotor layer close to the turbines (Figure iv).

[Figure]

*Figure viii: Time-averaged vertical velocity at the bottom (z = 27 m) of the turbine rotor layer for each atmospheric condition. The vertical velocity is averaged in the y-direction over the span of the wind plant.*

We extend our discussion on shear as follows:

Line 300: The vertical velocity advects horizontal momentum out of the turbine rotor layer (Figure 19). Vertical advection of horizontal momentum is 20% larger in the moderate stability case compared to the weak stability upstream of the first turbine row (Figure 19b). Larger vertical shear of the horizontal velocity in the moderate stability case compared to the weak stability case is the primary cause for the increased vertical advection of horizontal momentum. Shear $\left(\frac{\Delta \overline{u}}{\Delta z} = \frac{\overline{u}_t - \overline{u}_b}{D}\right)$ between the bottom $\overline{u}_b$ and top $\overline{u}_t$ of the turbine rotor layer is 43.6% larger in the -0.5 K/h simulation compared to the -0.2 K/h simulation. Similarly, the vertical velocity in the turbine rotor layer is 20% larger in the moderate stability case than in the weak stability case between x = -6D and x = 0D. The vertical velocity is expected to be larger in the moderate stability case because, as shown in Figure 18, the streamwise slowdown of the flow is transformed almost entirely into vertical motions. As a result, advection of horizontal momentum by the vertical velocity $\Delta(\rho \overline{u} \, \overline{w} \, S_z) = \rho S_z (\overline{u}_t \overline{w}_t - \overline{u}_b \overline{w}_b)$ is amplified."

We include the vertical velocity upstream of a front-row turbine in the middle of the wind plant and of a stand-alone turbine as follows:

Line 309: "Therefore, differences in vertical transport of horizontal momentum between a stand-alone turbine and a turbine in the wind plant are entirely due to the vertical velocity that forms upstream of the turbine array (Figure 21). For the wind plant, the secondary flow (i.e., net upwards w-velocity) extends farther upstream than for a stand-alone turbine (Figure 21)."

[Figure]

Figure 21: Streamwise evolution of the vertical velocity upstream of a stand-alone and front-row turbine in middle of the wind plant. The vertical velocity is averaged in the y-direction over the rotor diameter and in the z-direction over the top half of the rotor layer.

C. This is a big statement in the paper: "Given that the normalized pressure gradient force remains unchanged with atmoshperic stability and turbine array size, differences in blockage are caused by momentum redistribution in the induction region." Firstly, you'll want to correct the misspelling of atmospheric. Secondly, I just want to put this into context. In practice, when evaluating the impact of blockage or wakes on a wind farm, what we care about is the power production of the wind farm turbines relative to what they each would produce in isolation. In other words, we care about the wind conditions that each turbine experiences relative to the conditions it would experience in isolation. My interpretation of your analysis is that, at least for the simulated conditions, the vertical advection of u- momentum is *the dominant factor* affecting the production of leading row turbines relative to what they would produce in isolation. It is by far the main physical mechanism behind blockage loss for these turbines—again, for the simulated conditions. Am I interpreting your work correctly?

Thank you for highlighting the spelling mistake, we corrected the manuscript accordingly. Your interpretation is correct. We also extend our analysis on the power production of the turbines compared to a stand-alone turbine as follows:

Line 195: "Even though the wind speed slowdown from blockage is small, front-row turbines in the wind plant produce on average 5.2% less power than a stand-alone turbine (Figure 10). Because winds are slightly faster in the moderate stability case compared to the weak stability case, turbine power is also expected to differ. As a result, we evaluate the difference in power production between the turbines in the wind plant and a stand-alone turbine for the same atmospheric conditions. Just as the velocity deceleration is modified with atmospheric stability, turbine underperformance is more severe in the moderate stability case compared to the weak stability case. Whereas turbines in the first row produce on average 4% less power than a stand-alone turbine for the weak stability condition, front-row turbines produce on average 6.5% less power than a stand-alone in the moderate stability case. Downstream of the first row of the wind plant, turbine power is primarily dominated by the evolution of the wake. Turbine wakes persist longer in stable boundary layers because of reduced turbulence mixing (Dörenkämper et al., 2015; Lee and Lundquist, 2017), so we expect downstream turbines to produce less power in the moderate stability case compared to the weak stability case."

[Figure]

Figure 10: Normalized turbine power for each row of the wind plant and each atmospheric condition. The mean turbine power for the i-th row of the wind plant $\overline{P_i}$ is normalized over the mean turbine power of a stand-alone turbine $\overline{P_{st}}$.

D. I'm not sure your simulation setup can reliably capture gravity waves. With our own steady- state RANS model, a domain height of 3500 m (too low) and a damping layer thickness of 1000 m (too thin) would yield significantly different results with respect to gravity waves than our standard domain (top boundary at 17,000 m and much thick damping layer). That said, if you were to re-run your analysis with a much larger domain, I doubt your main findings would be much different. Thus, in my view, it is not required to run this sensitivity check, though it would be a nice-to-have. If future studies focus on gravity waves—and more broadly the influence of the stably stratified atmosphere above the boundary layer—such a sensitivity study would be needed.

Thank you for your suggestion, we agree that this analysis would be very helpful. However, the computational resources required for this sensitivity analysis are not available to us at this moment so we cannot include this in our manuscript.

E. As suggested above, I would consider just dropping Appendix C. However, if you keep it, could you please clarify the height at which the values in Figure C1 are being plotted?

Thank you for highlighting missing information. We decided to keep the Appendix on gravity waves because the other two reviewers consider this information important for our analysis and discussion.

We update the caption of Figure C1 to include the height for the data as follows:

Figure C1: "Streamwise evolution of the vertical velocity, pressure and potential temperature deviation from inflow conditions for the weak (a) and moderate (b) stability cases at z = 1200 m. Each variable $a_i$ is normalized as $\widehat{a}_i = \frac{a_i}{\max(a_i)-\min(a_i)}$ and averaged along the y-direction (from y = 1953 m to y = 5922 m). The gray shaded area in each panel represents the region covered by the wind plant."

F. If you pursue this research further, it would be interesting to know what you find when simulating a neutral boundary layer and/or an unstable boundary layer.

Thank you for your recommendations. We agree that this would be an interesting and useful path for future research.